# A crosstalk between hepcidin and IRE/IRP pathways controls ferroportin expression and determines serum iron levels in mice

Edouard Charlebois[1], Carine Fillebeen[1], Angeliki Katsarou[1], Aleksandr Rabinovich[2], Kazimierz Wisniewski[2], Vivek Venkataramani[3,4], Bernhard Michalke[5], Anastasia Velentza[2], Kostas Pantopoulos[1]*

[1]Lady Davis Institute for Medical Research, Jewish General Hospital and Department of Medicine, McGill University, Montreal, Canada; [2]Ferring Research Institute Inc, San Diego, United States; [3]Department of Medicine II, Hematology/Oncology, University Hospital Frankfurt, Frankfurt, Germany; [4]Institute of Pathology, University Medical Center Göttingen (UMG), Göttingen, Germany; [5]Helmholtz Zentrum München GmbH – German Research Center for Environmental Health, Research Unit Analytical BioGeoChemistry, Neuherberg, Germany

**Abstract** The iron hormone hepcidin is transcriptionally activated by iron or inflammation via distinct, partially overlapping pathways. We addressed how iron affects inflammatory hepcidin levels and the ensuing hypoferremic response. Dietary iron overload did not mitigate hepcidin induction in lipopolysaccharide (LPS)-treated wild type mice but prevented effective inflammatory hypoferremia. Likewise, LPS modestly decreased serum iron in hepcidin-deficient $Hjv^{-/-}$ mice, model of hemochromatosis. Synthetic hepcidin triggered hypoferremia in control but not iron-loaded wild type animals. Furthermore, it dramatically decreased hepatic and splenic ferroportin in $Hjv^{-/-}$ mice on standard or iron-deficient diet, but only triggered hypoferremia in the latter. Mechanistically, iron antagonized hepcidin responsiveness by inactivating IRPs in the liver and spleen to stimulate ferroportin mRNA translation. Prolonged LPS treatment eliminated ferroportin mRNA and permitted hepcidin-mediated hypoferremia in iron-loaded mice. Thus, de novo ferroportin synthesis is a critical determinant of serum iron and finetunes hepcidin-dependent functional outcomes. Our data uncover a crosstalk between hepcidin and IRE/IRP systems that controls tissue ferroportin expression and determines serum iron levels. Moreover, they suggest that hepcidin supplementation therapy is more efficient when combined with iron depletion.

## Editor's evaluation

The authors present a manuscript aiming to understand how systemic iron overload counteracts the hypoferremic effects of a specific inflammatory stimulus, specifically focused on the role of mechanisms of ferroportin regulation to achieve hypoferremia during inflammation. This work is of interest to the community of researchers interested in the interaction of systemic iron regulation and inflammation and possibly ultimately clinicians managing iron disorders. This work also is of novel significance for translational purposes and could lead to the design of better therapeutics for iron related disorders and/or anemia of chronic inflammation. The current study demonstrates LPS and exogenous hepcidin can synergistically lead to hypoferremia even in iron overload conditions and provides data implicating ferroportin translation in contributing to the fully sequestering iron in cells involved in iron flows to induce hypoferremia.

**\*For correspondence:**
kostas.pantopoulos@mcgill.ca

## Introduction

Systemic iron balance is controlled by hepcidin, a peptide hormone that is produced by hepatocytes in the liver and operates in target cells by binding to the iron exporter ferroportin (*Camaschella et al., 2020*; *Katsarou and Pantopoulos, 2020*). This results in ferroportin internalization and lysosomal degradation but also directly inhibits ferroportin function by occluding its iron export channel (*Aschemeyer et al., 2018*; *Billesbølle et al., 2020*). Ferroportin is highly expressed in duodenal enterocytes and tissue macrophages, which are instrumental for dietary iron absorption and iron recycling from senescent erythrocytes, respectively. Ferroportin is also expressed in hepatocytes, where excess iron is stored and can be mobilized on demand. Hepcidin-mediated ferroportin inactivation inhibits iron entry into plasma. This is a critical homeostatic response against iron overload, but also an innate immune response against infection (*Ganz and Nemeth, 2015*). Thus, hepcidin expression is induced when systemic iron levels are high to prevent dietary iron absorption or under inflammatory conditions to promote iron retention within ferroportin-expressing cells and render the metal unavailable to extracellular pathogens.

The hepcidin-encoding *Hamp* gene is transcriptionally induced by iron or inflammatory stimuli via BMP/SMAD (*Wang and Babitt, 2019*) or IL-6/STAT3 (*Schmidt, 2015*) signaling, respectively. These pathways crosstalk at different levels. For instance, the BMP co-receptor hemojuvelin (HJV), a potent enhancer of iron-dependent BMP/SMAD signaling, is also essential for the inflammatory induction of hepcidin. Thus, *Hjv$^{-/-}$* mice, a model of juvenile hemochromatosis characterized by severe iron overload and hepcidin deficiency (*Huang et al., 2005*), exhibit blunted inflammatory induction of hepcidin and fail to mount a hypoferremic response following LPS treatment or infection with *E. coli* (*Fillebeen et al., 2018*). Excess iron inhibits hepcidin induction via the BMP/SMAD and IL-6/STAT3 signaling pathways in cultured cells (*Charlebois and Pantopoulos, 2021*; *Yu et al., 2021*), but the in vivo relevance of these findings is not known.

Hepcidin-dependent inhibition of ferroportin activity and expression is a major but not the sole contributor to inflammatory hypoferremia (*Guida et al., 2015*; *Deschemin and Vaulont, 2013*). This is related to the fact that ferroportin expression is regulated by additional transcriptional and post-transcriptional mechanisms (*Drakesmith et al., 2015*). Thus, ferroportin transcription is induced by iron (*Aydemir et al., 2009*) and suppressed by inflammatory signals (*Ludwiczek et al., 2003*), while translation of *Slc40a1(+IRE)* mRNA, the major ferroportin transcript that harbors an 'iron responsive element' (IRE) within its 5' untranslated regions (5' UTR) is controlled by 'iron regulatory proteins' (IRPs), IRP1 and IRP2. The IRE/IRP system accounts for coordinate post-transcriptional regulation of iron metabolism proteins in cells (*Wang and Pantopoulos, 2011*; *Muckenthaler et al., 2008*). In a homeostatic response to iron deficiency, IRPs bind to the IRE within the *Slc40a1(+IRE)* and ferritin (*Fth1* and *Ftl1*) mRNAs, inhibiting their translation. IRE/IRP interactions do not take place in iron-loaded cells, allowing de novo ferroportin and ferritin synthesis to promote iron efflux and storage, respectively. The impact of the IRE/IRP system on the regulation of tissue ferroportin and serum iron is not well understood.

The aim of this work was to elucidate mechanisms by which systemic iron overload affects hepcidin expression and downstream responses, especially under inflammatory conditions. Utilizing wild type and *Hjv$^{-/-}$* mice, we demonstrate that serum iron levels reflect regulation of ferroportin in the liver and spleen by multiple signals. We further show that effective hepcidin-mediated hypoferremia is antagonized by compensatory mechanisms aiming to prevent cellular iron overload. Our data uncovered a crosstalk between hepcidin and the IRE/IRP system that controls ferroportin expression in the liver and spleen, and thereby determines serum iron levels.

## Results

### Dietary iron overload does not prevent further inflammatory *Hamp* mRNA induction in LPS-treated wild type mice, but mitigates hepcidin responsiveness

In an exploratory experiment, wild type mice were subjected to dietary iron loading by feeding a high-iron diet (HID) for short (1 day), intermediate (1 week), or long (5 weeks) time intervals; other animals remained on standard (control) diet. As expected, mice on HID for 1 day manifested maximal increases in serum iron (*Figure 1A*) and transferrin saturation (*Figure 1B*). They retained physiological liver iron

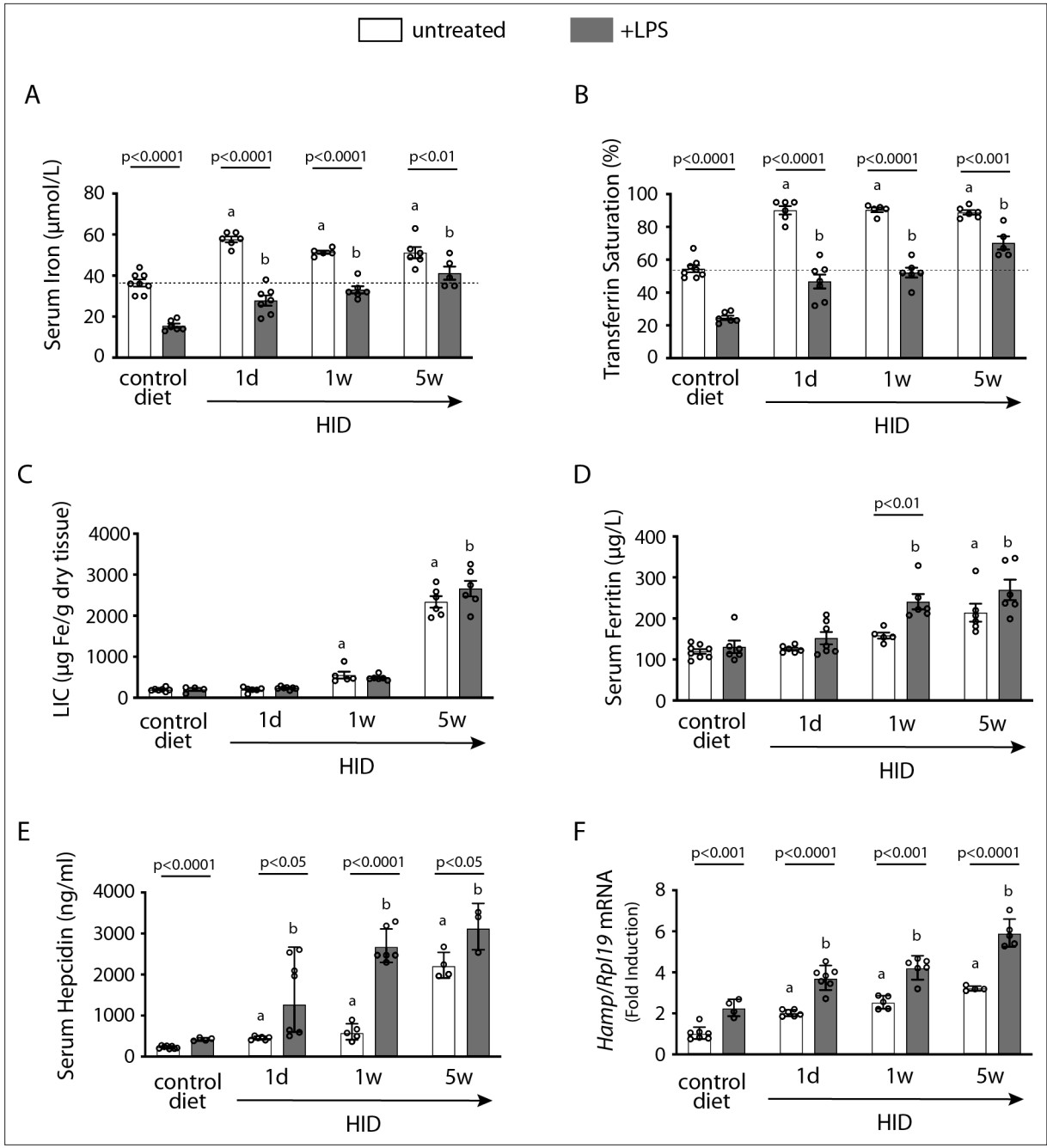

**Figure 1.** Dietary iron loading does not disrupt inflammatory hepcidin induction in LPS-treated wild type mice but blunts hepcidin-mediated hypoferremia. Nine-week-old male mice (n=12–14 per group) were fed control diet or high-iron diet (HID) for 1 day, 1 week, or 5 weeks prior to sacrifice. Half of the mice were injected intraperitoneally with saline and the other half with 1 µg/g LPS 4 hr before sacrifice. Sera were collected by cardiac puncture and analyzed for: (**A**) iron, (**B**) transferrin saturation, (**D**) ferritin, and (**E**) hepcidin. Livers were dissected and processed for biochemical analysis of: (**C**) liver iron content (LIC) by the ferrozine assay and (**F**) *Hamp* mRNA by qPCR. The dotted line in (**A**) and (**B**) indicates baseline serum iron and transferrin saturation, respectively, from mice on control diet. Data (**A–E**) are presented as the mean ± SEM and in (**F**) as geometric mean ± SD. Statistically significant differences (p<0.05) over time compared to values from saline- or LPS-treated control mice are indicated by a or b, respectively.

The online version of this article includes the following source data for figure 1:

**Source data 1.** qPCR data.

**Source data 2.** Serum hepcidin calculations.

**Source data 3.** Liver iron quantification.

**Source data 4.** Serum iron and transferrin saturation values.

content (LIC; [*Figure 1C*]) and serum ferritin (*Figure 1D*), a reflection of LIC. Serum iron and transferrin saturation plateaued after longer HID intake, while LIC and serum ferritin gradually increased to peak at 5 weeks. The dietary iron loading promoted gradual upregulation of serum hepcidin (*Figure 1E*) and liver *Hamp* mRNA (*Figure 1F*), with highest values at 5 weeks. This could not prevent chronic dietary iron overload, in agreement with earlier findings (*Corradini et al., 2011*; *Daba et al., 2013*).

LPS triggered appropriate hepcidin induction and a robust hypoferremic response in control mice. Interestingly, LPS-induced inflammation resulted in further proportional increase in hepcidin and *Hamp* mRNA in dietary iron-loaded mice (*Figure 1E–F*). This was accompanied by significant drops in serum iron and transferrin saturation (*Figure 1A–B*). However, values did not reach the nadir of LPS-treated control animals and were increasing in mice on HID for longer periods, despite significant hepcidin accumulation. These data suggest that hepatic iron overload does not prevent inflammatory induction of hepcidin; however, it impairs its capacity to decrease serum iron.

## Uncoupling inflammatory hepcidin induction from hypoferremic response in wild type and *Hjv*[-/-] mice following dietary iron manipulations

To further explore the potential of hepcidin to promote hypoferremia under iron overload, wild type and *Hjv*[-/-] mice, a model of hemochromatosis, were subjected to dietary iron manipulations. Wild type mice were fed control diet or HID, and *Hjv*[-/-] mice were fed control diet or an iron-deficient diet (IDD) for 5 weeks, to achieve a broad spectrum of hepcidin regulation. Wild type mice on HID and *Hjv*[-/-] mice on control diet or IDD manifested similarly high serum iron and transferrin saturation (*Figure 2A–B*). Serum non-transferrin bound iron (NTBI) levels appeared modestly elevated in the dietary and genetic iron overload models and seemed to decrease in *Hjv*[-/-] mice following IDD intake (*Figure 2C*). LIC was substantially reduced in *Hjv*[-/-] mice in response to IDD but also compared to wildtype mice on HID (*Figure 2D*). The quantitative LIC data were corroborated histologically by Perls staining (*Figure 2E* and *Figure 2—figure supplement 1A*). Dietary iron loading increased splenic iron in wild type mice and confirmed that *Hjv*[-/-] mice fail to retain iron in splenic macrophages (*Figure 2—figure supplement 1B*). As expected, serum hepcidin (*Figure 2F*) and liver *Hamp* mRNA (*Figure 2G*) were maximally induced in HID-fed wild type mice and were low in *Hjv*[-/-] mice on control diet, and further suppressed to undetectable levels following IDD intake.

LPS reduced serum iron and transferrin saturation in hyperferremic wild type mice on HID and *Hjv*[-/-] mice on control diet or IDD, but not below the baseline of wild type mice on control diet, the only animals that developed a robust hypoferremic response (*Figure 2A–B*); see also ratios of serum iron levels between untreated and LPS-treated mice in *Figure 2A*. The LPS treatment was associated with significant accumulation of hepcidin (*Figure 2F*) and induction of *Hamp* mRNA (*Figure 2G*) in all experimental groups, while NTBI (*Figure 2C*) and LIC (*Figure 2D*) were unaffected. Notably, LPS-treated wild type mice on HID and *Hjv*[-/-] mice on IDD exhibited dramatic differences in *Hamp* mRNA but similar blunted hypoferremic responses to the acute inflammatory stimulus. Thus, the profound hepcidin induction in iron-loaded wild type mice cannot decrease serum iron below that of iron-depleted *Hjv*[-/-] mice with negligible hepcidin, which indicates reduced hepcidin responsiveness. In support of this interpretation, *Id1* and *Socs3* mRNAs (*Figure 2H–I*), which are markers of BMP/SMAD and IL-6/STAT3 signaling, respectively, were appropriately induced by dietary iron loading or LPS treatment in wild type mice. Thus, the major hepcidin signaling pathways were intact under these experimental conditions.

Serum iron levels are also controlled by hepcidin-independent mechanisms (*Guida et al., 2015*; *Deschemin and Vaulont, 2013*). To explore their possible contribution in our experimental setting, we analyzed expression of genes involved in iron transport in the liver, an organ that contributes to iron sequestration during inflammation. Ferroportin is encoded by two alternatively spliced transcripts, *Slc40a1(+IRE)* and *Slc40a1(-IRE)* (*Zhang et al., 2009*). Both of them were significantly increased in the liver of iron-loaded wild type mice on HID and *Hjv*[-/-] mice on control diet, which is consistent with transcriptional induction (*Aydemir et al., 2009*), and were strongly suppressed by LPS (*Figure 2J–K*). The LPS treatment induced *Slc11a2*, *Slc39a14* and *Lcn2* mRNAs in all animals (*Figure 2L–N*). These encode the divalent metal transporter DMT1, the NTBI transporter Zip14 and the siderophore-binding protein Lcn2, respectively; *Lcn2* mRNA induction was dramatic. The transferrin receptor 1 (Tfr1)-encoding *Tfrc* mRNA was largely unaffected by LPS, except for a reduction in *Hjv*[-/-] mice on IDD (*Figure 2O*).

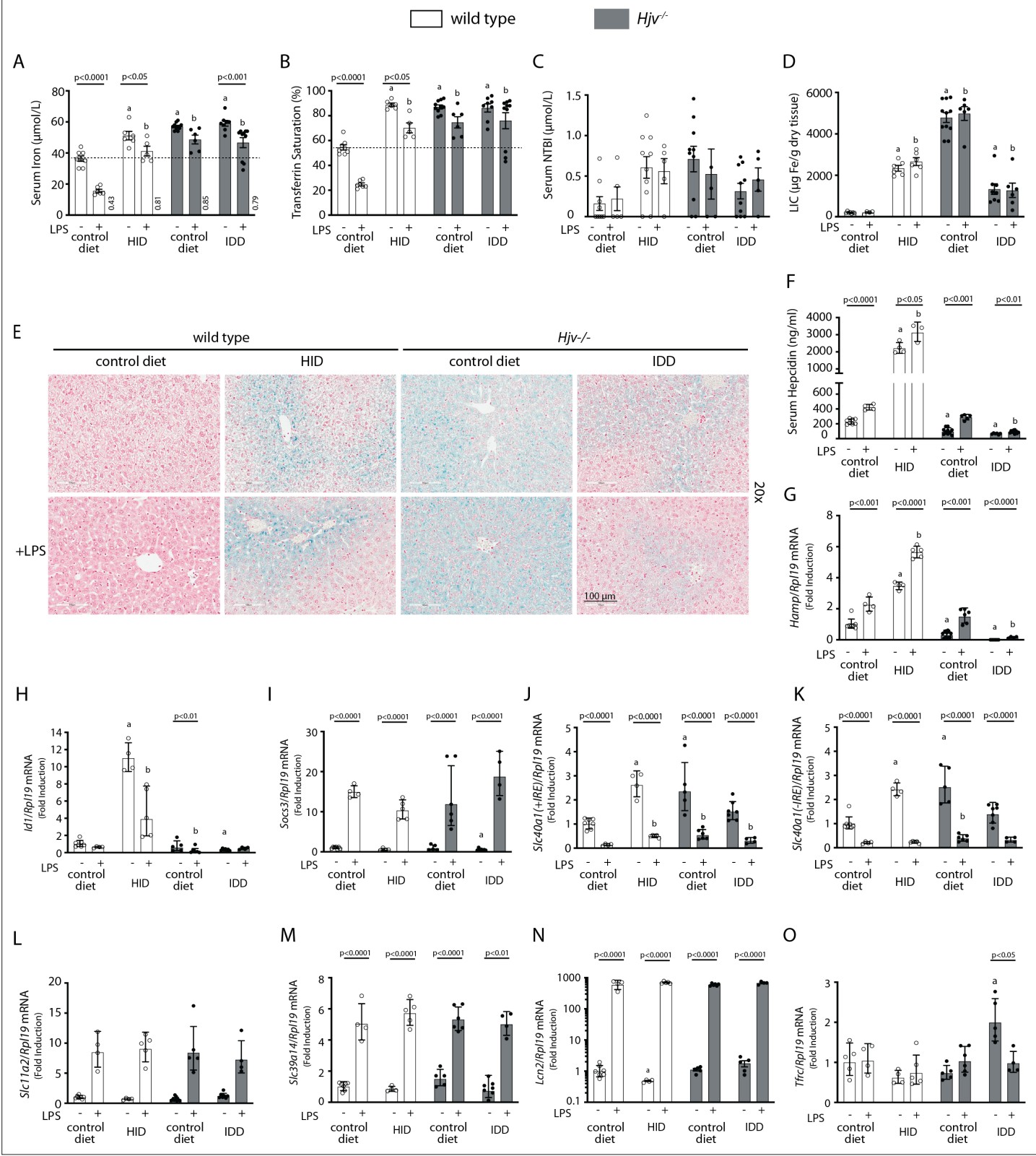

**Figure 2.** Iron overload blunts hepcidin responsiveness to LPS-induced inflammation. Four-week-old male wild type mice (n=12–14 per group) were placed on high-iron diet (HID) for 5 weeks. Conversely, age- and sex-matched isogenic *Hjv⁻/⁻* mice (n=12–14 per group) were placed on iron-deficient diet (IDD) for 5 weeks to prevent excessive iron overload. Other animals from both genotypes were kept on control diet. Half of the mice were injected with saline and the other half with 1 µg/g LPS; all animals were sacrificed 4 hr later. Sera were collected by cardiac puncture and analyzed for: (**A**) iron,

*Figure 2 continued on next page*

*Figure 2 continued*

(**B**) transferrin saturation, (**C**) non-transferrin bound iron (NTBI), and (**F**) hepcidin. Livers were dissected and processed for LIC quantification by the ferrozine assay (**D**) and for histological detection of iron deposits by Perls' staining (E; magnification: 20 ×). Livers were also used for qPCR analysis of following mRNAs: (**G**) *Hamp*, (**H**) *Id1*, (**I**) *Socs3*, (**J**) *Slc40a1(+IRE)*, (**K**) *Slc40a1(-IRE)*, (**L**) *Slc11a2*, (**M**) *Slc39a14*, (**N**) *Lcn2,* and (**O**) *Tfrc*. The dotted line in (**A**) and (**B**) indicates baseline serum iron and transferrin saturation, respectively, of wild type mice on control diet. Values in (**A**) represent ratios of serum iron levels between untreated and LPS-treated mice. Data in (**A–F**) are presented as the mean ± SEM while in (**G–O**) are presented as geometric mean ± SD. Statistically significant differences (p<0.05) compared to values from saline- or LPS-treated wild type control mice are indicated by a or b, respectively.

The online version of this article includes the following source data and figure supplement(s) for figure 2:

**Source data 1.** qPCR data.

**Source data 2.** Serum NTBI calculations.

**Source data 3.** Serum iron ratios.

**Source data 4.** Serum hepcidin calculations.

**Source data 5.** Liver iron quantification.

**Source data 6.** Serum iron and transferrin saturation values.

**Figure supplement 1.** Effects of dietary iron manipulations in hepatic and splenic iron of wild type and *Hjv⁻/⁻* mice.

The above data indicate that LPS-induced inflammation triggers transcriptional responses favoring reduced iron efflux from the liver and increased uptake of NTBI by liver cells.

To assess the downstream function of hepcidin, we analyzed tissue ferroportin levels. Immunohistochemical staining of liver sections revealed strong ferroportin expression in Kupffer cells, predominantly in periportal areas, under all experimental conditions (*Figure 3A* and *Figure 3—figure supplement 1*). Hepatocellular ferroportin staining is also evident in the iron overload models, mostly in periportal hepatocytes (*Figure 3—figure supplement 1*), and in line with recent data (*Katsarou et al., 2021*). LPS triggered redistribution and decreased expression of ferroportin in Kupffer cells from wild type but not *Hjv⁻/⁻* mice (*Figure 3—figure supplement 1*), as reported in *Fillebeen et al., 2018*.

We further analyzed ferroportin in liver homogenates by Western blotting. Levels of biochemically detectable liver ferroportin differed substantially between wild type and *Hjv⁻/⁻* mice. Thus, they were relatively low in the former and highly induced in the latter (*Figure 3B*), independently of iron load. The differences were more dramatic compared to those observed by immunohistochemistry (*Figure 3A* and *Figure 3—figure supplement 1*). Conceivably, the strong ferroportin signal in Western blots from *Hjv⁻/⁻* liver homogenates reflects high ferroportin expression in hepatocytes, which are the predominant cell population and make up ~80% of the liver cell mass (*Schulze et al., 2019*). Yet, hepatocellular ferroportin is less visible by immunohistochemistry because the signal is substantially weaker compared to that in Kupffer cells (see also Figure 6E). Interestingly, the LPS treatment visibly suppressed total liver ferroportin in *Hjv⁻/⁻* mice on control diet but not IDD, and appeared to modestly reduce it in wild type mice (*Figure 3B*); albeit without statistical significance. These data are consistent with negative regulation of ferroportin by residual LPS-induced hepcidin in *Hjv⁻/⁻* mice on control diet, which could explain the small drop in serum iron and transferrin saturation under these acute inflammatory conditions, as reported in *Fillebeen et al., 2018*. However, liver ferroportin remained detectable and apparently functional, as it did not allow significant iron sequestration and dramatic drop in serum iron. Notably, persistence of relatively high serum iron is also evident in LPS-treated wild type mice on HID, despite maximal hepcidin and minimal liver ferroportin levels.

Next, we analyzed ferroportin in the spleen, an organ with erythrophagocytic macrophages that plays an important role in body iron traffic (*Kurotaki et al., 2015*). Immunohistochemical analysis shows that LPS reduced ferroportin in red pulp splenic macrophages from wild type mice on control diet, but this effect was less evident in wild type mice on HID and in *Hjv⁻/⁻* mice on control diet or IDD (*Figure 3C* and *Figure 3—figure supplement 2*). Western blot analysis shows a stronger ferroportin signal in splenic extracts from *Hjv⁻/⁻* animals (*Figure 3D*), consistent with immunohistochemistry. However, in this assay, LPS suppressed splenic ferroportin in wild type animals and in *Hjv⁻/⁻* mice on control diet, but not IDD. This could be a result of residual hepcidin upregulation (*Figure 2F–G*), while the lack of significant splenic ferroportin suppression in *Hjv⁻/⁻* mice on IDD may denote hepcidin insufficiency. In any case, the relatively high circulating iron levels in dietary iron-loaded and LPS-treated wild type mice indicate continuous iron efflux to plasma despite hepcidin excess.

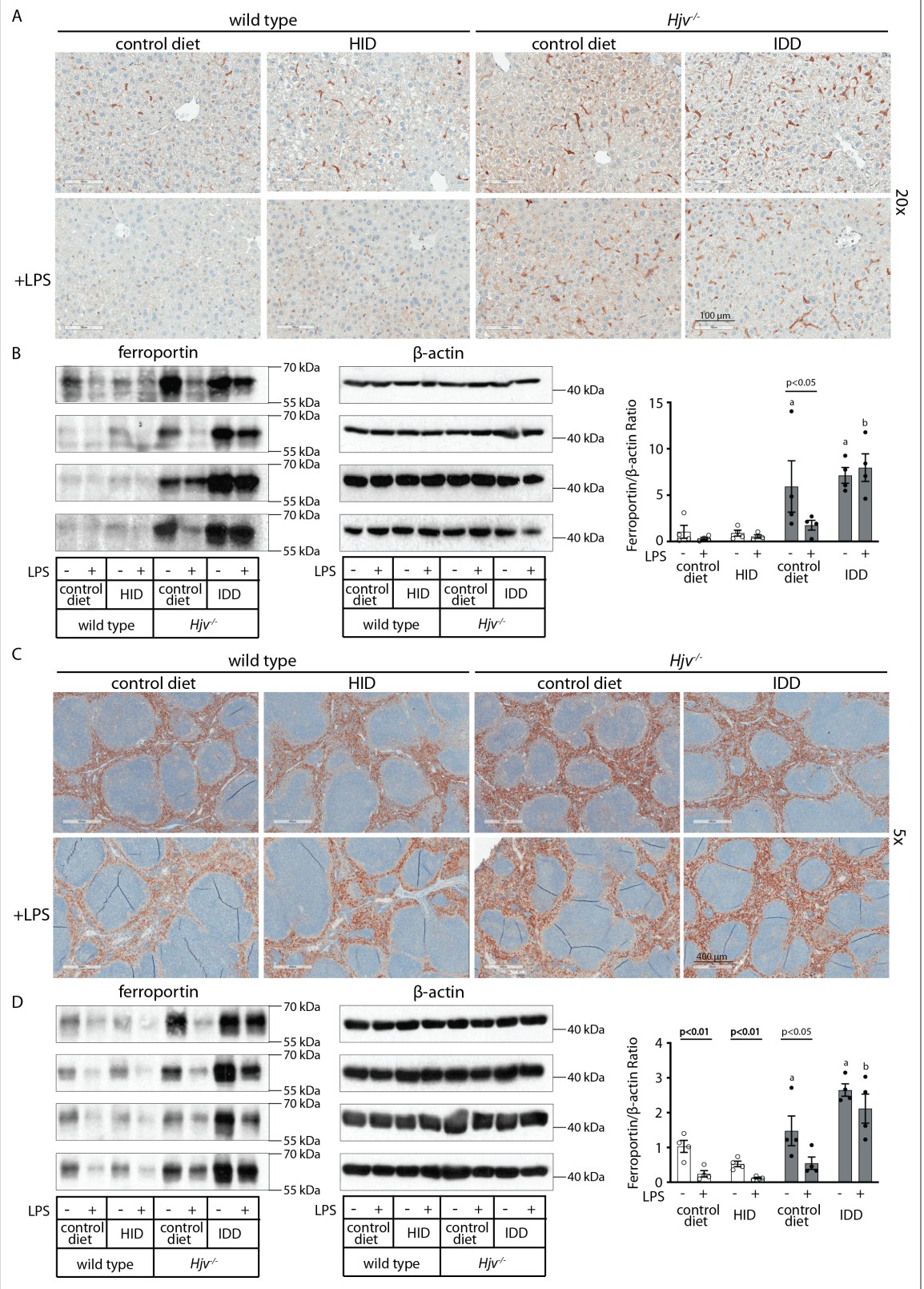

**Figure 3.** Effects of LPS on hepatic and splenic ferroportin of iron-manipulated wild type and *Hjv*−/− mice. Livers and spleens from mice described in *Figure 2* were dissected and processed for immunohistochemical and biochemical analysis of ferroportin. Immunohistochemical staining of ferroportin in liver (**A**) and spleen (**C**) sections (magnification for liver is 20 × and for spleen 5 ×). Western blot for ferroportin and β-actin in liver (**B**) and spleen (**D**) extracts from four representative mice in each condition. Blots were quantified by densitometry and ferroportin/β-actin ratios are shown on the right.

*Figure 3 continued on next page*

*Figure 3 continued*

Densitometric data are presented as the mean ± SEM. Statistically significant differences (p<0.05) compared to values from saline- or LPS-treated wild type control mice are indicated by a or b, respectively. Statistics in bold were performed using unpaired Student's t test. HID: high-iron diet; IDD: iron-deficient diet.

The online version of this article includes the following source data and figure supplement(s) for figure 3:

**Source data 1.** Western blot quantifications.

**Source data 2.** Raw unlabeled Western blot ferroportin *Figure 3D* (a).

**Source data 3.** Raw unlabeled Western blot ferroportin *Figure 3D* (b).

**Source data 4.** Raw unlabeled Western blot β-actin *Figure 3D*.

**Source data 5.** Raw unlabeled Western blot β-actin *Figure 3D* (b).

**Source data 6.** Raw unlabeled Western blot ferroportin *Figure 3B* (a).

**Source data 7.** Raw unlabeled Western blot β-actin *Figure 3B* (a).

**Source data 8.** Raw unlabeled Western blot ferroportin and β-actin *Figure 3B* (a).

**Source data 9.** Raw unlabeled Western blot ferroportin *Figure 3B* (b).

**Source data 10.** Raw unlabeled Western blot β-actin *Figure 3B* (b).

**Source data 11.** Raw labeled Western blot β-actin *Figure 3B* (b).

**Source data 12.** Raw labeled Western blot ferroportin *Figure 3B* (b).

**Source data 13.** Raw labeled Western blot ferroportin and β-actin *Figure 3B* (a).

**Source data 14.** Raw labeled Western blot ferroportin *Figure 3B* (a).

**Source data 15.** Raw labeled Western blot β-actin *Figure 3B* (a).

**Source data 16.** Raw labeled Western blot ferroportin *Figure 3D* (b).

**Source data 17.** Raw labeled Western blot ferroportin *Figure 3D* (a).

**Source data 18.** Raw labeled Western blot β-actin *Figure 3D* (b).

**Source data 19.** Raw labeled Western blot β-actin *Figure 3D* (a).

**Figure supplement 1.** Low magnification immunohistochemical images of ferroportin in liver sections of dietary iron-manipulated wild type and *Hjv*[-/-] mice following LPS treatment.

**Figure supplement 2.** Low magnification immunohistochemical images of ferroportin in spleen sections of dietary iron-manipulated wild type and *Hjv*[-/-] mice following LPS treatment.

## Insufficient hepcidin leads to blunted hypoferremic response in iron overload

We used human synthetic hepcidin to address whether the failure of mouse models of iron overload to mount an appropriate hypoferremic response to acute inflammation is caused by endogenous hepcidin insufficiency or other mechanisms. Wild type and *Hjv*[-/-] mice subjected to dietary iron manipulations received 2.5 µg/g synthetic hepcidin every two hours for a total of four intraperitoneal injections. Each dose corresponds to ~200-fold excess over endogenous circulating hepcidin in wild type animals. The treatment caused hypoferremia in wild type mice on control diet but not on HID, where the decrease in serum iron was significant but well above baseline of untreated wild type controls (*Figure 4A–B*); see also ratios of serum iron levels between untreated and hepcidin-treated mice in *Figure 4A*. Likewise, synthetic hepcidin significantly decreased serum iron but failed to cause dramatic hypoferremia in hepcidin-deficient *Hjv*[-/-] mice on control diet. Notably, hepcidin administration was much more effective in relatively iron-depleted *Hjv*[-/-] mice on IDD, and lowered serum iron and transferrin saturation below baseline. The treatments significantly reduced NTBI in *Hjv*[-/-] mice on control diet, with a trend in mice on IDD (*Figure 4C*) but did not affect LIC or splenic iron content (SIC) under any experimental conditions (*Figure 4D–E* and *Figure 4—figure supplement 1*). Serum iron represents <2% of total tissue iron and, therefore, its acute fluctuations are not expected to dramatically alter LIC or SIC.

Synthetic hepcidin led to a significant reduction of endogenous *Hamp* mRNA in wild type mice on control diet (*Figure 4F*), as earlier reported (*Laftah et al., 2004*). Conceivably, this is related to destabilization of the Hamp inducer transferrin receptor 2 (Tfr2) in the liver (*Figure 4—figure supplement 2*), a known response to hypoferremia (*Johnson and Enns, 2004*). Synthetic hepcidin did not

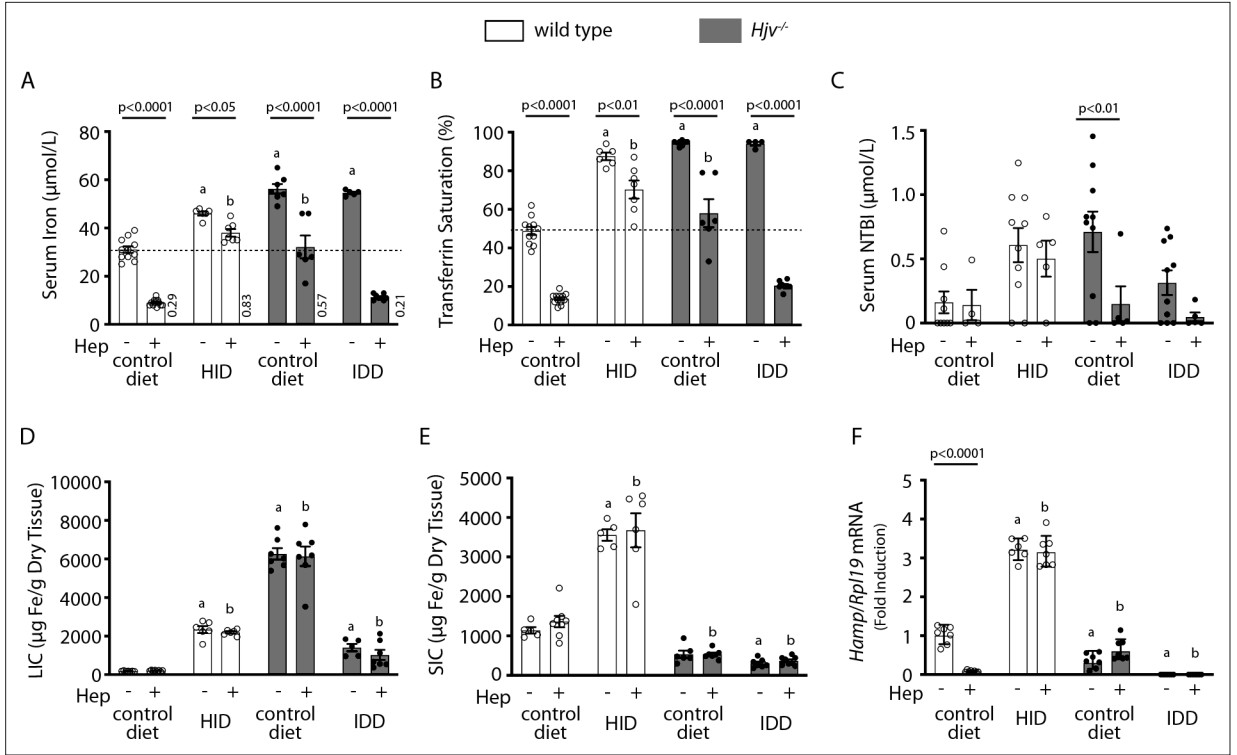

**Figure 4.** Iron depletion of *Hjv^-/-* mice improves the efficacy of synthetic hepcidin to promote hypoferremia. Four-week-old wild type male mice (n=12–14 per group) were placed on HID for 5 weeks. Conversely, age- and sex-matched isogenic *Hjv^-/-* mice (n=12–14 per group) were placed on IDD for 5 weeks to prevent excessive iron overload. Other animals from both genotypes were kept on standard diet. Half of the mice were injected every 2 hr for a total of 4 injections with saline, and the other half with 2.5 μg/g synthetic hepcidin. Sera were collected by cardiac puncture and analyzed for: (**A**) iron, (**B**) transferrin saturation, and (**C**) non-transferrin bound iron (NTBI). Livers and spleens were dissected and processed for analysis of: (**D**) liver iron content (LIC) and (**E**) splenic iron content (SIC) by the ferrozine assay. (**F**) qPCR analysis of liver *Hamp* mRNA. The dotted line in (**A**) and (**B**) indicates baseline serum iron and transferrin saturation, respectively, of wild type mice on control diet. Values in (**A**) represent ratios of serum iron levels between untreated and hepcidin-treated mice. Data in (**A–E**) are presented as the mean ± SEM and in (**F**) as geometric mean ± SD. Statistically significant differences (p<0.05) compared to values from saline- or hepcidin-treated wild type control mice are indicated by a or b, respectively. HID: high-iron diet; IDD: iron-deficient diet.

The online version of this article includes the following source data and figure supplement(s) for figure 4:

**Source data 1.** qPCR data.

**Source data 2.** Liver and spleen iron quantification.

**Source data 3.** Serum NTBI calculations.

**Source data 4.** Serum hepcidin calculations.

**Source data 5.** Serum iron ratios.

**Source data 6.** Serum iron and transferrin saturation values.

**Figure supplement 1.** Perls staining for iron deposits in liver and spleen sections of dietary iron-manipulated wild type and *Hjv^-/-* mice following treatment with synthetic hepcidin.

**Figure supplement 2.** Western analysis of transferrin receptors (Tfr1 and Tfr2) of dietary iron-manipulated wild type and *Hjv^-/-* mice following treatment with synthetic hepcidin.

**Figure supplement 2—source data 1.** Western blot quantifications.

**Figure supplement 2—source data 2.** Raw unlabeled Western blot Tfr2 and β-actin.

**Figure supplement 2—source data 3.** Raw unlabeled Western blot Tfr1.

**Figure supplement 2—source data 4.** Raw labeled Western blot Tfr1.

**Figure supplement 2—source data 5.** Raw labeled Western blot Tfr2 and β-actin.

**Figure supplement 3.** Effects of LPS treatment on expression of mRNAs encoding iron transport proteins and signaling endpoints in the liver of dietary iron-manipulated wild type and *Hjv^-/-* mice.

**Figure supplement 3—source data 1.** qPCR data.

promote inflammation, iron perturbations or alterations in BMP/SMAD signaling in the liver, as judged by the unaltered expression of hepatic *Slc40a1(+IRE)*, *Socs3*, *Id1*, and *Bmp6* mRNAs (*Figure 4—figure supplement 3A-D*). Moreover, synthetic hepcidin did not affect *Slc11a2*, *Slc39a14*, *Lcn2*, or *Tfrc* mRNAs (*Figure 4—figure supplement 3E-H*), which encode iron transporters; *Slc39a14* and *Lcn2* are also inflammatory markers.

Next, we analyzed liver ferroportin by immunohistochemistry. *Figure 5A* and *Figure 5—figure supplement 1* show that exogenous hepcidin decreased ferroportin signal intensity in all animal groups to varying degrees. The hepcidin effect was particularly noticeable in *Hjv⁻/⁻* hepatocytes (see low magnification images in *Figure 5—figure supplement 1*). Kupffer cells seemed to retain some ferroportin in all groups except *Hjv⁻/⁻* mice on IDD. Interestingly, while synthetic hepcidin decreased ferroportin signal intensity in Kupffer cells, it did not alter intracellular ferroportin distribution as would be expected based on the data in LPS-treated wild type mice (*Figure 5A*).

Western blotting confirmed that total liver ferroportin is highly induced in *Hjv⁻/⁻* mice (*Figure 5B*). Again, the signal intensity can be attributed to proteins expressed in hepatocytes. The treatment with synthetic hepcidin did not significantly affect liver ferroportin in wild type mice (either on control diet or HID), but substantially reduced it in *Hjv⁻/⁻* mice, to almost wild type levels. The effect appeared more pronounced in *Hjv⁻/⁻* mice on IDD; nevertheless, ferroportin remained detectable.

Splenic ferroportin was reduced in all animal groups following hepcidin treatment, with stronger effects visualized by immunohistochemistry in wild type mice on control diet and *Hjv⁻/⁻* mice on IDD (*Figure 5C* and *Figure 5—figure supplement 2*). At the biochemical level, ferroportin expression was again much stronger in the spleen of *Hjv⁻/⁻* mice (*Figure 5D*). Synthetic hepcidin did not significantly affect splenic ferroportin in wild type mice but dramatically reduced it in all *Hjv⁻/⁻* mice.

Taken together, our data suggest that synthetic hepcidin overcomes endogenous hepcidin deficiency in *Hjv⁻/⁻* mice. However, it only triggers hypoferremia in these animals following relative iron depletion. On the other hand, in iron-loaded wild type mice with already high endogenous hepcidin, excess synthetic hepcidin fails to promote hypoferremia.

## Dietary iron manipulations are sensed by IRPs in the liver and spleen of wild type and *Hjv⁻/⁻* mice

The IRE/IRP system orchestrates homeostatic adaptation to cellular iron supply (*Wang and Pantopoulos, 2011*; *Muckenthaler et al., 2008*). To evaluate the responses of IRPs in the whole liver and spleen to the above-described dietary iron manipulations, we analyzed tissue extracts from wild type and *Hjv⁻/⁻* mice by an electrophoretic mobility shift assay (EMSA) using a ³²P-labelled IRE probe. The data in *Figure 6A–B* show that HID intake tended to decrease the IRE-binding activities of IRP1 and IRP2 in both the liver and spleen of wild type mice (statistical significance is only reached in the liver); densitometric quantification of IRE/IRP1 and IRE/IRP2 complexes is shown on the right. Conversely, IDD intake significantly induced the IRE-binding activity of IRP2 in the liver and spleen of *Hjv⁻/⁻* mice, leaving IRP1 largely unaffected. IRE/IRP2 interactions are better visible in longer exposures (middle panels). EMSAs with tissue extracts previously treated with 2-mercaptoethanol (2-ME) were performed as loading controls (*Fillebeen et al., 2014*) and are shown in the bottom panels.

To clarify which cell types of the liver account for the responses of IRPs to dietary iron, separate EMSAs were performed using extracts from isolated hepatocytes or non-parenchymal liver cells. The data in *Figure 6C–D* uncover that IRP1 and IRP2 in both liver cell populations from wild type and *Hjv⁻/⁻* mice are sensitive to dietary iron loading or restriction, respectively. The EMSA analysis of non-parenchymal liver cells, which contain Kupffer cells among others, showed a high experimental variability (*Figure 6D*). Nevertheless, the overall results are consistent with those obtained with splenic extracts, which contain red pulp macrophages (*Figure 6B*).

## Relative expression of ferroportin in hepatocytes and non-parenchymal liver cells from wild type and *Hjv⁻/⁻* mice

We determined the relative abundance of ferroportin in hepatocytes and non-parenchymal liver cells from wild type and *Hjv⁻/⁻* mice on control diet by Western blotting. As expected, ferroportin expression (normalized to β-actin) was ~1.5–twofold higher in the non-parenchymal cell fraction as compared to hepatocytes in both wild type and *Hjv⁻/⁻* mice (*Figure 6E*). In comparison across genotypes, ferroportin

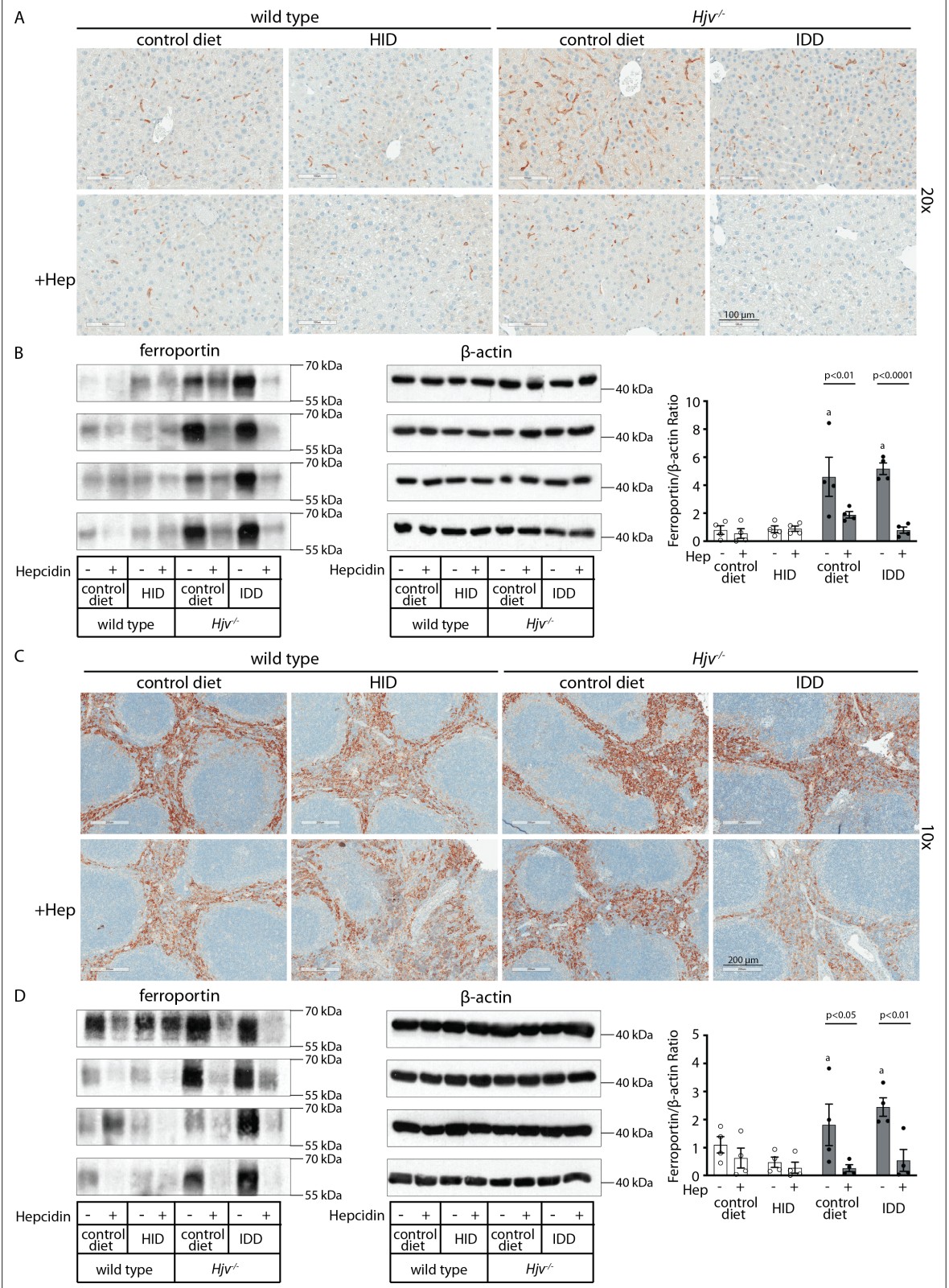

**Figure 5.** Effects of synthetic hepcidin on hepatic and splenic ferroportin of iron-manipulatedwild type and *Hjv⁻/⁻* mice. Livers and spleens from mice described in *Figure 4* were dissected and processed for immunohistochemical and biochemical analysis of ferroportin. Immunohistochemical staining of ferroportin in liver (**A**) and spleen (**C**) sections (magnification for liver is 20 × and for spleen 10 ×). Western blot for ferroportin and β-actin in liver (**B**) and spleen (**D**) extracts from four representative mice in each condition. Blots were quantified by densitometry and ferroportin/β-actin ratios are

*Figure 5 continued on next page*

Figure 5 continued

shown on the right. Densitometric data are presented as the mean ± SEM. Statistically significant differences (p<0.05) compared to values from saline- or hepcidin-treatedwild type control mice are indicated by a or b, respectively. HID: high-iron diet; IDD: iron-deficient diet.

The online version of this article includes the following source data and figure supplement(s) for figure 5:

**Source data 1.** Western blot quantifications.

**Source data 2.** Raw unlabeled Western blot ferroportin *Figure 5D* (a).

**Source data 3.** Raw unlabeled Western blot β-actin *Figure 5D* (a).

**Source data 4.** Raw unlabeled Western blot ferroportin *Figure 5B* (a).

**Source data 5.** Raw unlabeled Western blot ferroportin *Figure 5B* (b).

**Source data 6.** Raw unlabeled Western blot ferroportin *Figure 5D* (b).

**Source data 7.** Raw unlabeled Western blot.

**Source data 8.** Raw unlabeled Western blot ferroportin *Figure 5D* (c).

**Source data 9.** Raw unlabeled Western blot β-actin *Figure 5D* (c).

**Source data 10.** Raw unlabeled Western blot β-actin *Figure 5B* (a).

**Source data 11.** Raw unlabeled Western blot β-actin *Figure 5B* (b).

**Source data 12.** Raw labeled Western blot β-actin *Figure 5B* (b).

**Source data 13.** Raw labeled Western blot β-actin *Figure 5B* (a).

**Source data 14.** Raw labeled Western blot ferroportin *Figure 5B* (b).

**Source data 15.** Raw labeled Western blot ferroportin *Figure 5B* (a).

**Source data 16.** Raw labeled Western blot β-actin *Figure 5D* (a).

**Source data 17.** Raw labeled Western blot ferroportin *Figure 5D* (a).

**Source data 18.** Raw labeled Western blot β-actin *Figure 5D* (c).

**Source data 19.** Raw labeled Western blot ferroportin *Figure 5D* (c).

**Source data 20.** Raw labeled Western blot β-actin *Figure 5D* (b).

**Source data 21.** Raw labeled Western blot ferroportin *Figure 5D* (b).

**Source data 22.** Raw labeled Western blot ferroportin and β-actin *Figure 5D* (a).

**Source data 23.** Raw unlabeled Western blot ferroportin and β-actin *Figure 5D* (a).

**Figure supplement 1.** Low magnification immunohistochemical images of ferroportin in liver sections of dietary iron-manipulated wild type and $Hjv^{-/-}$ mice following treatment with synthetic hepcidin.

**Figure supplement 2.** Low magnification immunohistochemical images of ferroportin in spleen sections of dietary iron-manipulated wild type and $Hjv^{-/-}$ mice following treatment with synthetic hepcidin.

expression was ~2-fold higher in hepatocytes and ~50% higher in non-parenchymal cells from $Hjv^{-/-}$ vs wild type mice.

## Iron-dependent regulation of ferroportin mRNA translation in the liver

Having established that dietary iron manipulations trigger IRP responses in the liver and spleen, we hypothesized that the functional outcomes of exogenous hepcidin may not merely depend on its capacity to degrade tissue ferroportin but also on iron-dependent ferroportin regeneration via de novo synthesis. *Slc40a1(+IRE)* mRNA is the predominant ferroportin transcript in the mouse liver and spleen, as well as in hepatoma and macrophage cell lines (*Zhang et al., 2009*), and is considered as a target of IRPs.

Thus, we assessed the effects of dietary iron on whole liver *Slc40a1(+IRE)* mRNA translation by polysome profile analysis. We focused on the liver because this organ contains the highest number of iron-recycling macrophages (*Krenkel and Tacke, 2017*) and can also export iron to plasma from ferroportin-expressing parenchymal cells. Liver extracts from wild type mice on control diet or HID, and $Hjv^{-/-}$ mice on control diet or IDD were fractionated on sucrose gradients to separate translationally inactive light monosomes from translating heavy polysomes (*Figure 7A*). The relative distribution of *Slc40a1(+IRE)*, *Fth1* (positive control for iron regulation), and *Actb* (negative control) mRNAs within the different fractions was quantified by qPCR (*Figure 7B–D*). Dietary iron loading stimulated *Slc40a1(+IRE)* (and *Fth1*) mRNA translation in wild type mice (note the shifts from monosomes to

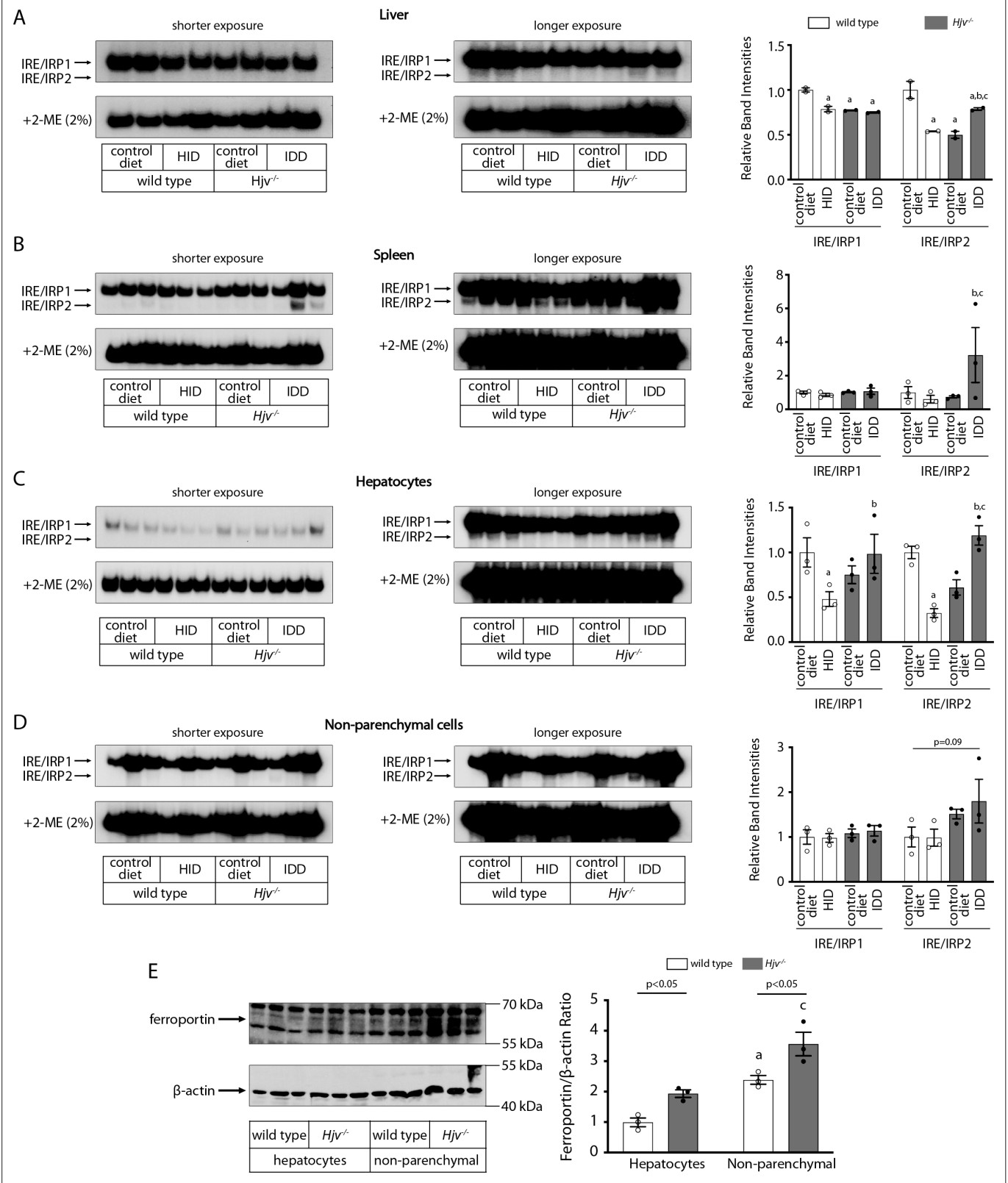

**Figure 6.** Dietary iron manipulations trigger IRP responses in the liver and spleen, as well as in primary hepatocytes and non-parenchymal liver cells of wild type and *Hjv⁻/⁻* mice. Whole liver (**A**), whole spleen (**B**), isolated hepatocytes (**C**) or isolated non-parenchymal liver cells (**D**) from the mice described in *Figure 4* were analyzed for IRE-binding activity by EMSA with a ³²P-labelled IRE probe in the absence (top) or presence (bottom) of 2% mercaptoethanol (2-ME). Two or three representative samples from each condition are shown. The positions of IRE/IRP1 and IRE/IRP2 complexes are

*Figure 6 continued on next page*

*Figure 6 continued*

indicated by arrows. Shorter and longer exposures of the autoradiograms are shown in the left and middle panels, respectively. Relative band intensities were quantified by densitometry and shown on the right panels. (**E**) Isolated hepatocytes and isolated non-parenchymal liver cells were analyzed by Western blotting for expression of ferroportin and β-actin. Blots were quantified by densitometry and ferroportin/β-actin ratios are shown on the right. Densitometric data are presented as the mean ± SEM. Statistically significant differences (p<0.05) in values from wild type mice on control diet are indicated by a, from wild type mice on HID by b, and from *Hjv*[-/-] mice on control diet by c. HID: high-iron diet; IDD: iron-deficient diet; IRE: iron-responsive element; IRP: iron regulatory protein; EMSA: electrophoretic mobility shift assay.

The online version of this article includes the following source data for figure 6:

**Source data 1.** EMSA quantification.

**Source data 2.** Western quantification.

**Source data 3.** Raw unlabeled liver EMSA long exposure *Figure 6A*.

**Source data 4.** Raw unlabeled liver EMSA long exposure 2-ME *Figure 6A*.

**Source data 5.** Raw unlabeled liver EMSA short exposure *Figure 6A*.

**Source data 6.** Raw unlabeled liver EMSA short exposure 2-ME *Figure 6A*.

**Source data 7.** Raw unlabeled spleen EMSA long exposure *Figure 6B*.

**Source data 8.** Raw unlabeled spleen EMSA long exposure 2-ME *Figure 6B*.

**Source data 9.** Raw unlabeled spleen EMSA short exposure *Figure 6B*.

**Source data 10.** Raw unlabeled spleen EMSA short exposure BME *Figure 6B*.

**Source data 11.** Raw unlabeled Western blot β-actin *Figure 6E*.

**Source data 12.** Raw unlabeled Western blot ferroportin *Figure 6E*.

**Source data 13.** Raw unlabeled hepatocytes EMSA short exposure *Figure 6C*.

**Source data 14.** Raw unlabeled hepatocytes EMSA short exposure 2-ME *Figure 6C*.

**Source data 15.** Raw unlabeled hepatocytes EMSA long exposure *Figure 6C*.

**Source data 16.** Raw unlabeled hepatocytes EMSA long exposure 2-ME *Figure 6C*.

**Source data 17.** Raw unlabeled non-parenchymal cells EMSA short exposure *Figure 6D*.

**Source data 18.** Raw unlabeled non-parenchymal cells EMSA short exposure 2-ME *Figure 6D*.

**Source data 19.** Raw unlabeled non-parenchymal cells EMSA long exposure *Figure 6D*.

**Source data 20.** Raw unlabeled non-parenchymal cells EMSA long exposure 2-ME *Figure 6D*.

**Source data 21.** Raw labeled liver EMSA long exposure *Figure 6A*.

**Source data 22.** Raw labeled liver EMSA long exposure 2-ME *Figure 6A*.

**Source data 23.** Raw labeled liver EMSA short exposure *Figure 6A*.

**Source data 24.** Raw labeled liver EMSA short exposure 2-ME *Figure 6A*.

**Source data 25.** Raw labeled spleen EMSA short exposure *Figure 6B*.

**Source data 26.** Raw labeled spleen EMSA short exposure 2-ME *Figure 6B*.

**Source data 27.** Raw labeled spleen EMSA long exposure *Figure 6B*.

**Source data 28.** Raw labeled spleen EMSA long exposure 2-ME *Figure 6B*.

**Source data 29.** Raw labeled hepatocytes EMSA short exposure *Figure 6C*.

**Source data 30.** Raw labeled hepatocytes EMSA short exposure 2-ME *Figure 6C*.

**Source data 31.** Raw labeled hepatocytes EMSA long exposure *Figure 6C*.

**Source data 32.** Raw labeled hepatocytes EMSA long exposure 2-ME *Figure 6C*.

**Source data 33.** Raw labeled non-parenchymal cells EMSA short exposure *Figure 6D*.

**Source data 34.** Raw labeled non-parenchymal cells EMSA short exposure 2-ME *Figure 6D*.

**Source data 35.** Raw labeled non-parenchymal cells EMSA long exposure *Figure 6D*.

**Source data 36.** Raw labeled non-parenchymal cells EMSA long exposure 2-ME *Figure 6D*.

**Source data 37.** Raw labeled Western blot ferroportin and β-actin *Figure 6E*.

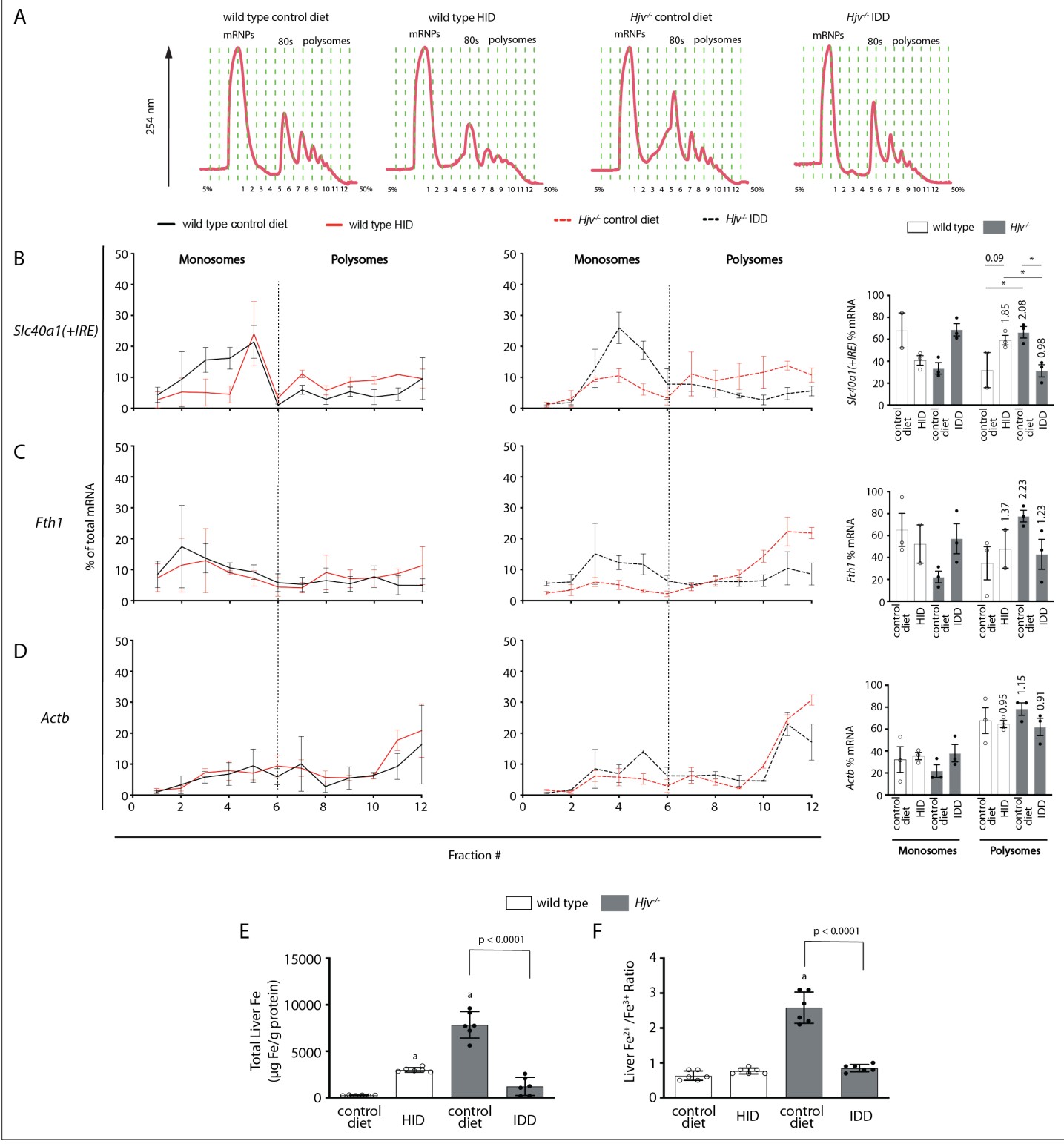

**Figure 7.** Iron regulation of *Slc40a1(+IRE)* mRNA translation in the mouse liver. Four-week-old wild type male mice (n=10–14 per group) were placed on high-iron diet (HID) for 5 weeks. Conversely, age- and sex-matched isogenic *Hjv⁻/⁻* mice (n=10–14 per group) were placed on iron-deficient diet (IDD) for 5 weeks to prevent excessive iron overload. Other animals from both genotypes were kept on control diet. At the endpoint, the mice were sacrificed, and livers were used for polysome profile analysis and iron assays. (**A**) Recording of absorbance at 254 nm of representative samples. Fraction numbers and direction of the gradient are indicated. (**B–D**) Liver polysome profiles from n=3 mice in each experimental group. Distribution of (**B**) *Slc40a1(+IRE)*, (**C**) *Fth1* and (**D**) *Actb* mRNAs among light monosomal and heavy polysomal fractions (separated by dashed line) was analyzed by qPCR. Bar graph

*Figure 7 continued on next page*

Figure 7 continued

comparisons of pooled fractions are shown on the right. Numbers indicate the fold change compared towild type mice oncontrol diet. (**E and F**) Analysis of total iron (**E**), and redox iron speciation (**F**) in the liver by CE-ICP-MS. Data are presented as the mean ± SEM. Statistical analysis in (**A**) was performed by two-way ANOVA and in (**B, C**) by one-way ANOVA. Statistically significant differences (p<0.05) compared to values from wild type mice on control are indicated by a.

The online version of this article includes the following source data for figure 7:

**Source data 1.** Polysome raw data.

**Source data 2.** Iron ratio calculations.

polysomes in *Figure 7B–C*). Conversely, dietary iron depletion inhibited *Slc40a1(+IRE)* (and *Fth1*) mRNA translation in *Hjv$^{-/-}$* mice. We also attempted to obtain polysome profiles of *Slc40a1(-IRE)* mRNA but it was undetectable after fractionation. These data indicate that in mice subjected to iron overload, iron-stimulated ferroportin synthesis in the liver antagonizes hepcidin-mediated ferro-portin degradation and prevents an appropriate hypoferremic response. Considering that levels of *Slc40a1(+IRE)* mRNA are elevated in iron-loaded wild type and *Hjv$^{-/-}$* mice (*Figure 2J* and *Figure 4—figure supplement 3*), it is possible that increased de novo ferroportin synthesis is further enhanced by transcriptional induction.

Quantification of liver iron by ICP-MS (*Figure 7E*) validated iron loading of wild type mice by HID, and relative iron depletion of *Hjv$^{-/-}$* mice by IDD intake, respectively (see also *Figure 2D*). Iron redox speciation analysis by CE-ICP-MS revealed a profound increase in Fe$^{2+}$/Fe$^{3+}$ ratios in livers of *Hjv$^{-/-}$* mice on control diet, which was corrected by dietary iron depletion (*Figure 7F*). Nevertheless, there was no difference in Fe$^{2+}$/Fe$^{3+}$ ratios among the livers of wild type mice on control diet or HID, and *Hjv$^{-/-}$* mice on IDD. We conclude that a relative increase in total iron content, rather than excessive accumulation of redox active Fe$^{2+}$ drives *Slc40a1(+IRE)* (and *Fth1*) mRNA translation in the liver.

## Restoration of effective hypoferremic response under iron overload following maximal *Slc40a1* mRNA suppression

We reasoned that complete inactivation of ferroportin mRNA would restore hepcidin-induced hypo-ferremia despite iron overload. An 8 hr treatment of mice with LPS suppressed liver *Slc40a1(+IRE)* mRNA below detection levels (*Figure 8A*), as reported (*Fillebeen et al., 2018*). The same holds true for the *Slc40a1(-IRE)* isoform (*Figure 8B*), which was 290 times less abundant in control mouse livers compared to *Slc40a1(+IRE)* (ΔCt = 8.18), in agreement with published data (*Zhang et al., 2009*). We went on to examine the effects of synthetic hepcidin on serum iron under these conditions of maximal *Slc40a1* mRNA suppression. Importantly, the prolonged LPS treatment decreased serum iron in wild type mice on HID below the control baseline (*Figure 8C*). Furthermore, when combined with synthetic hepcidin, it promoted an effective hypoferremic response in wild type mice on HID and *Hjv$^{-/-}$* mice on control diet (or IDD; *Figure 8C–D*) and tended to decrease NTBI (*Figure 8E*). These data strongly suggest that the expression of actively translating *Slc40a1* mRNA in iron-exporting tissues under systemic iron overload mitigates the hepcidin-induced drop in serum iron.

## Discussion

We sought to analyze how iron overload affects hepcidin-mediated inflammatory responses. We and others reported that excess iron inhibits the major hepcidin signaling pathways (BMP/SMAD and IL-6/STAT3) in cultured cells (*Charlebois and Pantopoulos, 2021*; *Yu et al., 2021*). To explore the phys-iological relevance of these findings, wild type mice were subjected to variable degrees of dietary iron loading and then treated with LPS. All iron-loaded mice could further upregulate hepcidin in response to LPS-induced acute inflammation (*Figure 1*). This is consistent with other relevant find-ings (*Enculescu et al., 2017*) and apparently contradicts the in vitro data. While experimental iron loading of cultured cells is rapid, dietary iron loading of mice is gradual (*Daba et al., 2013*) and most of the excess iron is effectively detoxified within ferritin, which is highly induced (*Kent et al., 2015*). By contrast, the suppression of hepcidin preceded ferritin induction in cultured cells (*Charlebois and Pantopoulos, 2021*), which may explain the discrepancy with the in vivo data.

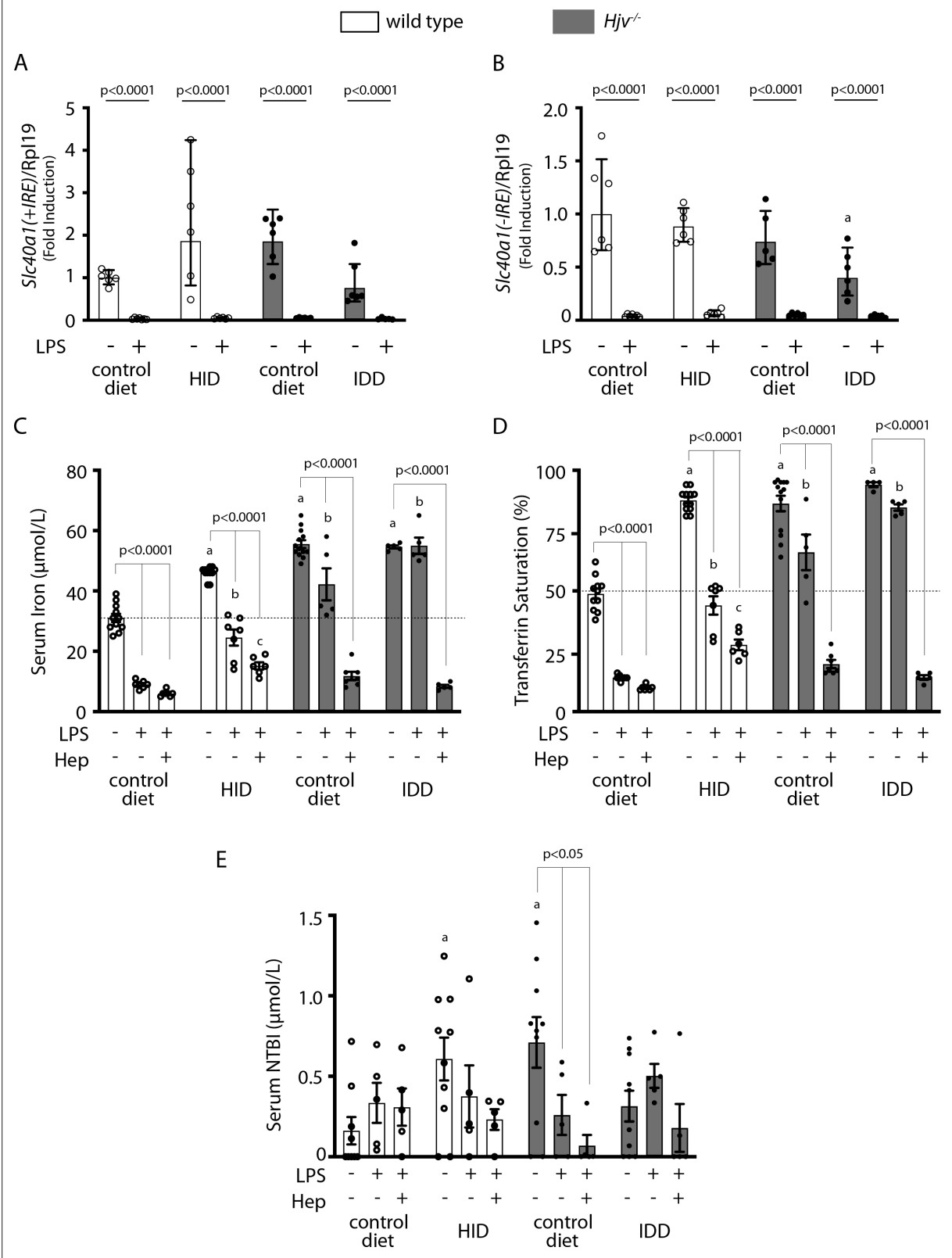

**Figure 8.** Elimination of ferroportin mRNA by prolonged LPS treatment potentiates hepcidin-induced hypoferremia in mouse models of iron overload. Four-week-old wild type male mice (n=10–14 per group) were placed on high-iron diet (HID) for 5 weeks. Conversely, age- and sex-matched isogenic *Hjv*⁻/⁻ mice (n=10–14 per group) were placed on iron-deficient diet (IDD) for 5 weeks to prevent excessive iron overload. Other animals from both genotypes were kept on control diet. (**A and B**) Half of the mice were injected with saline and the other half with 1 µg/g LPS and sacrificed after 8 hr.

*Figure 8 continued on next page*

*Figure 8 continued*

Livers were dissected and processed for qPCR analysis of *Slc40a1(+IRE)* (**A**) and *Slc40a1(-IRE)* (**B**) mRNAs. (**C–E**) All mice were injected with 1 µg/g LPS. Half of the animals were subsequently injected with saline, and the other half with 2.5 µg/g synthetic hepcidin every 2 hr for a total of 4 injections. At the endpoint the mice were sacrificed. Sera were collected by cardiac puncture and analyzed for: (**C**) iron, (**D**) transferrin saturation, and (**E**) non-transferrin bound iron (NTBI). The dotted line in (**C**) and (**D**) indicates baseline serum iron and transferrin saturation, respectively, of wild type mice on control diet. Data are presented as (**A–B**) geometric mean ± SD or (**C–E**) mean ± SEM. Statistically significant differences (p<0.05) compared to values from saline-, LPS- or hepcidin-treated wild type mice on control diet are indicated by a, b or c, respectively.

The online version of this article includes the following source data for figure 8:

**Source data 1.** qPCR data.

**Source data 2.** Serum NTBI calculations.

**Source data 3.** Serum iron and transferrin saturation values.

The unimpaired inflammatory induction of hepcidin in iron-loaded wild type mice correlated with significant drops in serum iron, but these appeared inversely proportional to the degree of systemic iron loading (*Figure 1*). Thus, LPS-treated mice on 5 weeks of HID developed relative hypoferremia but could not further reduce serum iron below the baseline of untreated mice on control diet. This can be attributed to mechanisms antagonizing hepcidin action. To explore how iron modulates the capacity of hepcidin to trigger inflammatory hypoferremia, we established conditions of iron overload using wild type and *Hjv⁻ᐟ⁻* mice with extreme differences in hepcidin expression. *Figures 2 and 3* demonstrate that iron overload prevents effective inflammatory hypoferremia independently of hepcidin and tissue ferroportin levels.

We used a~200-fold excess of synthetic hepcidin to directly assess its capacity to divert iron traffic in iron-loaded mice. Hepcidin injection caused hypoferremia in wild type mice on control diet and significantly reduced serum iron in wild type mice on HID and *Hjv⁻ᐟ⁻* mice on control diet, but not below baseline (*Figure 4*). Thus, synthetic hepcidin failed to drastically drop serum iron levels in iron over-load models with either high or low endogenous hepcidin. Importantly, synthetic hepcidin promoted robust hypoferremia in relatively iron-depleted *Hjv⁻ᐟ⁻* mice on IDD, with undetectable endogenous hepcidin. It should be noted that synthetic hepcidin had similar effects on tissue ferroportin among wild type or *Hjv⁻ᐟ⁻* mice, regardless of iron diet (*Figure 5*). It reduced the intensity of the ferroportin signal in Kupffer cells and splenic macrophages of wild type mice without significantly affecting biochemically detectable total protein levels. In addition, it dramatically reduced the total ferroportin in the liver and spleen of *Hjv⁻ᐟ⁻* mice. However, in all experimental settings, there was residual tissue ferroportin, which appears to be functionally significant.

We reasoned that at steady-state, tissue ferroportin may consist of fractions of newly synthesized protein and protein that is en route to hepcidin-mediated degradation. Conceivably, the former may exhibit more robust iron export activity, at least before its iron channel gets occluded by hepcidin. Increased de novo synthesis of active ferroportin could explain why synthetic hepcidin cannot drastically drop serum iron levels under iron overload. In fact, *Figure 7* demonstrates that dietary iron overload augments *Slc40a1(+IRE)* mRNA translation in the liver of wild type mice. Conversely, relative dietary iron depletion inhibits *Slc40a1(+IRE)* mRNA translation in the liver of *Hjv⁻ᐟ⁻* mice, in line with the restoration of hepcidin-mediated hypoferremic response (*Figure 4*).

Our data are consistent with translational control of liver ferroportin expression via the IRE/IRP system and do not exclude the possibility for an additional contribution of iron-dependent transcriptional regulation of *Slc40a1(+IRE)* mRNA. Direct evidence for activation of IRP responses in the liver and spleen to dietary iron manipulations is provided in *Figure 6*. While translational control of ferritin in tissues is established (*Wilkinson and Pantopoulos, 2014*), regulation of ferroportin by the IRE/IRP system is less well characterized and has hitherto only been documented in cell models (*Lymboussaki et al., 2003*; *Nairz et al., 2015*), the mouse duodenum (*Galy et al., 2013*), and the rat liver (*Garza et al., 2020*). Moreover, the physiological relevance of this mechanism remained speculative. The data in *Figures 6 and 7* show that the IRE/IRP system is operational and controls *Slc40a1(+IRE)* mRNA translation in both fractions of hepatocytes and non-parenchymal liver cells. Presumably, this offers a compensatory mechanism to protect the cells from iron overload and iron-induced toxicity. On the other hand, this mechanism attenuates hepcidin responsiveness and promotes a state of hepcidin resistance, as higher amounts of hepcidin are required to achieve effective hypoferremia. Because hepcidin has a short plasma half-life, it is reasonable to predict that the use of more potent hepcidin

analogs (*Katsarou and Pantopoulos, 2018*) will overcome the antagonistic effects of increased ferroportin mRNA translation under iron overload.

The critical role of de novo ferroportin synthesis in fine-tuning hepcidin-dependent functional outcomes is also highlighted in *Figure 8*. Thus, synthetic hepcidin was highly effective as a promoter of hypoferremia in dietary iron-loaded wild type mice when administered together with LPS. LPS is known to suppress *Slc40a1* mRNA in cell lines (*Ludwiczek et al., 2003*) and mouse tissues, with a nadir in the liver reached at 8 hr (*Fillebeen et al., 2018*). The recovery of hepcidin effectiveness in mouse models of iron overload was only possible when *Slc40a1* mRNA was essentially eliminated. Under these conditions, LPS treatment alone was sufficient to decrease serum iron in dietary iron-loaded wild type mice below baseline.

Tissue iron uptake may be another important determinant of the hypoferremic response to inflammation. LPS did not affect *Tfrc* mRNA levels in the liver (*Figure 2O*), which argues against increased uptake of transferrin-bound iron via Tfr1. On the other hand, LPS induced *Slc39a14*, *Slc11a2*, and *Lcn2* mRNAs (*Figure 2L–N*). Zip14 is the NTBI transporter accounting for hepatocellular iron overload in hemochromatosis (*Jenkitkasemwong et al., 2015*) and is upregulated by inflammatory cues in hepatocytes (*Liuzzi et al., 2005*). DMT1 is dispensable for NTBI uptake by hepatocytes (*Wang and Knutson, 2013*), and its inflammatory induction might promote iron acquisition by macrophages (*Ludwiczek et al., 2003*; *Wardrop and Richardson, 2000*). Nevertheless, considering that the fraction of NTBI represents <2% of total serum iron even in the iron overload models (*Figure 2A and C*), it is implausible that NTBI uptake by Zip14 and/or DMT1 substantially contributes to inflammatory hypoferremia. Lcn2 is an acute phase protein that can sequester intracellular iron bound to catechole siderophores (*Xiao et al., 2017*), and is more likely to transport iron to tissues during infection. In any case, synthetic hepcidin did not affect expression of iron transporters (*Figure 4—figure supplement Figure 4—figure supplement 3E-H*). This excludes the possibility for a synergistic effect on LPS-induced tissue iron uptake that could promote effective hypoferremia in the iron overload models.

Our study has some limitations. While the data highlight the importance of translational regulation of liver ferroportin as a determinant of serum iron, they do not accurately dissect the specific role of ferroportin expressed in hepatocytes and Kupffer cells; the latter were not separated from other non-parenchymal cells in biochemical assays. The involvement of the IRE/IRP system has been established indirectly, while the relative contributions of IRP1 and IRP2 in the mechanism are not fully defined. The possible role of iron-dependent transcriptional induction of ferroportin in counterbalancing hepcidin actions requires further clarification. The use of diets with variable iron content may have triggered responses to iron availability independent of hepcidin signaling and Hjv functionality. Finally, the physiological implications of translational regulation of ferroportin in the broader setting of inflammation and/or infection have not been explored.

In conclusion, our data reveal a crosstalk between the hepcidin pathway and the IRE/IRP system in the liver and spleen for the control of tissue ferroportin and serum iron levels. Furthermore, they suggest that application of hepcidin therapeutics for treatment of iron overload disorders should be combined with iron depletion strategies to mitigate *Slc40a1* mRNA translation and increase hepcidin efficacy. Future work is expected to clarify whether optimizing the hypoferremic response to inflammation under systemic iron overload decreases susceptibility to pathogens.

## Materials and methods

### Animals

Wild type C57BL/6 J and isogenic $Hjv^{-/-}$ mice (*Gkouvatsos et al., 2014*) were housed in macrolone cages (up to 5 mice/cage, 12:12 hr light-dark cycle: 7 am–7 pm; 22 ± 1°C, 60 ± 5% humidity). The mice were fed either a standard control diet (200 ppm iron), an iron-deficient diet (2–6 ppm iron) or a high-iron diet (2% carbonyl iron) (*Fillebeen et al., 2019*). Where indicated, mice were injected intraperitoneally with 1 µg/g LPS (serotype 055:B5; Sigma-Aldrich) or subcutaneously with 2.5 µg/g synthetic hepcidin; control mice were injected with phosphate-buffered saline. At the endpoints, animals were sacrificed by $CO_2$ inhalation and cervical dislocation. Experimental procedures were approved by the Animal Care Committee of McGill University (protocol 4966).

## Serum biochemistry

Blood was collected via cardiac puncture. Serum was prepared by using micro Z-gel tubes with clotting activator (Sarstedt) and was kept frozen at −20°C until analysis. Serum iron, total iron binding capacity (TIBC) and, where indicated serum ferritin, were determined at the Biochemistry Department of the Montreal Jewish General Hospital using a Roche Hitachi 917 Chemistry Analyzer. Transferrin saturation was calculated from the ratio of serum iron and TIBC. Serum hepcidin was measured by using an ELISA kit (HMC-001; Intrinsic LifeSciences).

## Quantification of serum non-transferrin bound iron (NTBI)

NTBI was measured by adapting the method developed by Esposito et a*l* (*Esposito et al., 2003*). Briefly, iron samples of known concentration were created by mixing 70 mM nitrilotriacetate (NTA) (pH = 7.0) with 20 mM ferrous ammonium sulfate. $Fe^{2+}$ was allowed to oxidize to $Fe^{3+}$ in ambient air for at least 30 min and then the solution was diluted to 0.2 mM before further serial dilutions to create a ladder. 5 µl of ladder was loaded in a 96-well plate containing 195 µl plasma-like medium with or without 100 µM deferiprone. The composition of the plasma-like medium was: 40 mg/ml bovine serum albumin, 1.2 mM sodium phosphate dibasic, 120 µM sodium citrate, 10 mM sodium bicarbonate in iron-free HEPES-buffered saline (HEPES 20 mM, NaCl 150 mM, treated with Chelex-100 chelating resin [Bio-Rad, Hercules, CA], 0.5 mM NTA, 40 µM ascorbic acid, 50 µM dihydrorhodamine, pH = 7.4). 5 µl of sample was loaded in a 96-well plate containing 195 µl of iron-free HEPES-buffered saline with or without 100 µM deferiprone. Microplates were read every 2 min at 37°C over 40 min at 485/520 nm (ex/em). Final NTBI was calculated by comparing the oxidation rate of DHR in the presence or absence of the strong chelator deferiprone.

## Hepcidin synthesis

Human hepcidin (DTHFPICIFCCGCCHRSKCGMCCKT) was synthesized at Ferring Research Institute, San Diego, CA. The linear peptide was assembled on Rink amide resin using Tribute peptide synthesizer and the peptide was cleaved from the resin with the TFA/TIS/EDT/$H_2O$ 91:3:3:3 (v/v/v/v) cocktail. The solvents were evaporated, and the crude peptide was precipitated with diethyl ether, reconstituted in 50% aqueous acetonitrile and lyophilized. The lyophilizate was dissolved in 30% aqueous acetonitrile at the concentration of 0.05 mM and the pH of the solution was adjusted to 7.8 with 6 M ammonium hydroxide. Folding was achieved within 4 hr using the cysteine/cystine redox (peptide/$Cys/Cys_2$ 1:6:6 molar ratio). The reaction mixture was acidified to pH 3, loaded onto HPLC prep column and purified in a TFA based gradient. The identity of the peptide was confirmed by mass spectrometry and by coelution with a commercially available sample (Peptide International, #PLP-3771-PI).

## Quantitative real-time PCR (qPCR)

RNA was extracted from livers by using the RNeasy kit (Qiagen). cDNA was synthesized from 1 µg RNA by using the OneScript Plus cDNA Synthesis Kit (Applied Biological Materials Inc). Gene-specific primers pairs (*Supplementary file 1*) were validated by dissociation curve analysis and demonstrated amplification efficiency between 90–110%. SYBR Green (Bioline) and primers were used to amplify products under following cycling conditions: initial denaturation 95°C 10 min, 40 cycles of 95°C 5 s, 58°C 30 s, 72°C 10 s, and final cycle melt analysis between 58–95°C. Relative mRNA expression was calculated by the $2^{-\Delta\Delta Ct}$ method (*Livak and Schmittgen, 2001*). Data were normalized to murine ribosomal protein L19 (*Rpl19*). Data are reported as fold increases compared to samples from wild type mice on control diet.

## Polysome fractionation

RNA was freshly prepared from frozen livers. Linear sucrose gradients were prepared the day before the experiment by using 5% (w/v) and 50% (w/v) sucrose solutions with 10 × gradient buffer (200 mM HEPES pH = 7.6, 1 M KCl, 50 mM $MgCl_2$, 0.1 mg/ml Cycloheximide, 1 tablet cOmplete, Mini, EDTA-free Protease Inhibitor Cocktail (Roche), 200 U/mL Recombinant RNasin Ribonuclease Inhibitor (Promega), 2 mM DTT). Linear gradients were prepared in Polyallomer Centrifuge Tubes (Beckman Coulter). Tubes were marked using a gradient cylinder (BioComp), and 5% sucrose solution was added using a syringe with a layering needle (BioComp) until solution level reached the mark. Then, 50% sucrose solution was layered underneath the 5% solution until the interface between the two solutions reached the

mark. Tubes were capped with rate zonal caps (BioComp) and linearized using a Gradient Master 108 (Biocomp). All reagents were nuclease-free and all solutions were kept on ice or at 4°C. Sample preparation was adapted from *Liang et al., 2018*. Briefly, livers were flash frozen upon collection. Roughly 30–80 mg of tissue was crushed using a mortar and pestle in the presence of liquid nitrogen to prevent thawing. Tissues were lysed in up to 1 ml of hypotonic lysis buffer (5 mM Tris-Hcl pH = 7.5, 1.5 mM KCl, 2.5 mM $MgCl_2$, 2 mM DTT, 1 mg/ml Cycloheximide, 200 U/ml Recombinant RNasin Ribonuclease Inhibitor [Promega], 1 tablet cOmplete, Mini, EDTA-free Protease Inhibitor Cocktail [Roche] 0.5% [v/v] Triton X-100, 0.5% [v/v] Sodium Deoxycholate) and homogenized using Dounce homogenizers (60 movements with both loose and tight pestles) on ice. Samples were centrifuged at 4°C, 16,060 g for 4 min and supernatants were collected. Sample optical density was measured at 260 nM and samples were normalized to either the lowest value or 30 ODs. 450 µl of sucrose gradient was removed from the top and replaced with normalized sample. Tube weights were balanced by weight before centrifugation at 200,000 g for 2 hr at 4°C in a SW 41 Ti rotor and a Beckman Optima L-60 Ultracentrifuge. Samples were fractionated using a BR-188 Density Gradient Fractionation System (Brandel). Immediately upon collection, 800 µl of samples were mixed with 1 ml of TRIzol and kept on ice before storage at –80°C. Polysomal RNA was processed according to the manufacturer's protocol. mRNA distribution was analyzed as previously described (*Panda et al., 2017*).

## Electrophoretic mobility shift assay (EMSA)

IRE-binding activities from liver and spleen were analyzed by EMSA using a radioactive [32]P-labelled IRE probe, according to established procedures (*Fillebeen et al., 2014*). EMSAs were also performed in extracts from hepatocytes and non-parenchymal cells, which were separated by using a 2-step collagenase perfusion technique, as previously described (*Fillebeen et al., 2018*).

## Western blotting

Livers were washed with ice-cold PBS and dissected into pieces. Aliquots were snap frozen at liquid nitrogen and stored at –80°C. Protein lysates were obtained as described (*Katsarou et al., 2021*). Lysates containing 40 µg of proteins were analyzed by SDS-PAGE on 9–13% gels and proteins were transferred onto nitrocellulose membranes (BioRad). The blots were blocked in non-fat milk diluted in tris-buffered saline (TBS) containing 0.1% (v/v) Tween-20 (TBS-T), and probed overnight with antibodies against ferroportin (*Ross et al., 2017*; 1:1000 diluted monoclonal rat anti-mouse 1C7, kindly provided by Amgen Inc), β-actin (1:2000 diluted; Sigma), Tfr2 (1:1000 diluted rabbit polyclonal; Alpha Diagnostics), or Tfr1 (1:1000 diluted mouse monoclonal, Invitrogen). Following a 3 × wash with TBS-T, the membranes were incubated with peroxidase-coupled secondary antibodies for 1 hr. Immunoreactive bands were detected by enhanced chemiluminescence with the Western Lightning ECL Kit (Perkin Elmer).

## Immunohistochemistry

Tissue specimens were fixed in 10% buffered formalin and embedded in paraffin. Samples from three different mice for each experimental condition were cut at 4 µm, placed on SuperFrost/Plus slides (Fisher) and dried overnight at 37°C. The slides were then loaded onto the Discovery XT Autostainer (Ventana Medical System) for automated immunohistochemistry. Slides underwent deparaffinization and heat-induced epitope retrieval. Immunostaining was performed by using 1:500 diluted rabbit polyclonal antibodies against ferroportin (*Maffettone et al., 2010*) and an appropriate detection kit (Omnimap rabbit polyclonal HRP, #760–4311 and ChromoMap-DAB #760–159; Roche). Negative controls were performed by the omission of the primary antibody. Slides were counterstained with hematoxylin for 4 min, blued with Bluing Reagent for 4 min, removed from the autostainer, washed in warm soapy water, dehydrated through graded alcohols, cleared in xylene, and mounted with Permount (Fisher). Sections were analyzed by conventional light microscopy and quantified by using the Aperio ImageScope software (Leica Biosystems; *Fillebeen et al., 2018*).

## Perls Prussian blue staining

To visualize non-heme iron deposits, deparaffinized tissue sections were stained with Perls' Prussian blue using the Accustain Iron Stain kit (Sigma).

## Quantification of liver iron content (LIC)

Total liver iron was quantified by using the ferrozine assay (*Daba et al., 2013*) or inductively coupled plasma mass spectrometry (ICP-MS; *Michalke et al., 2019*).

## Iron speciation analysis

Iron redox speciation analysis in the liver was performed by capillary electrophoresis (CE) coupled to ICP-MS (CE-ICP-MS). Dynamic reaction cell (DRC) technology (ICP-DRC-MS) with $NH_3$ as DRC-gas was applied for non-interfered monitoring of the iron isotopes. A 'PrinCe 706' CE system (PrinCe Technologies B.V., Emmen, Netherlands) was employed for separation of iron species at +20 kV. Temperature settings for sample/buffer tray and capillary were set to 20°C. An uncoated capillary (100 cm × 50 μm ID; CS-Chromatographie Service GmbH, Langerwehe, Germany) was used for separation and hyphenation to the ICP–DRC-MS. A CE-ICP-MS interface (*Michalke et al., 2019*; *Michalke et al., 2020*) was installed which provided the electrical connection between CE capillary end and outlet electrode. The self-aspiration mode allowed for best flow rate adjustment and avoided suction flow. Electrolytes for sample stacking and electrophoretic separation were 10% HCl = leading electrolyte, 0.05 mM HCl = terminating electrolyte and 50 mM HCl = background electrolyte. The $Fe^{2+}/Fe^{3+}$ ratio was calculated from quantitative determined concentrations of Fe-species.

## Statistics

Statistical analysis was performed by using the Prism GraphPad software (version 9.1.0). Lognormally distributed data including qPCR and ELISA results were first log transformed before analysis with ordinary two-way ANOVA (Tukey's multiple comparisons test) for comparisons within same treatment groups (denoted by a or b on figures) or with multiple unpaired t tests using the Holm-Sidak method to compare effects between treatments. Normally distributed data was analyzed by two-way ANOVA using either Sidak's method for comparisons between treatment groups or Tukey's multiple comparisons test within treatments groups. Where indicated, pairwise comparisons were done with unpaired Student's t test. Probability value $p < 0.05$ was considered statistically significant.

## Acknowledgements

We thank Dr. Naciba Benlimame and Lilian Canetti for assistance with histology and immunohistochemistry. This work was supported by a grant from the Canadian Institutes of Health Research (CIHR; PJT-159730). EC was funded by a fellowship from the Natural Sciences and Engineering Research Council of Canada (NSERC) and is currently a recipient of a fellowship from the *Fonds de recherche du Québec – Santé* (FRQS). The work of VV and BM was financially supported by the Deutsche Forschungsgemeinschaft (DFG) through the Priority Program "Ferroptosis: from Molecular Basics to Clinical Applications" (SPP 2306).

## Additional information

### Competing interests

Aleksandr Rabinovich, Kazimierz Wisniewski, Anastasia Velentza: Was an employee of Ferring Research Institute Inc. The other authors declare that no competing interests exist.

### Funding

| Funder | Grant reference number | Author |
|---|---|---|
| Canadian Institutes of Health Research | PJT-159730 | Kostas Pantopoulos |
| Fonds de Recherche du Québec - Santé | | Edouard Charlebois |
| Deutsche Forschungsgemeinschaft | SPP 2306 | Vivek Venkataramani |

| Funder | Grant reference number | Author |
| --- | --- | --- |

The funders had no role in study design, data collection and interpretation, or the decision to submit the work for publication.

## Author contributions

Edouard Charlebois, Investigation, Methodology; Carine Fillebeen, Investigation, Methodology, Project administration; Angeliki Katsarou, Investigation; Aleksandr Rabinovich, Kazimierz Wisniewski, Anastasia Velentza, Resources; Vivek Venkataramani, Bernhard Michalke, Methodology; Kostas Pantopoulos, Conceptualization, Supervision, Funding acquisition, Writing - original draft, Writing - review and editing

## Author ORCIDs
Kostas Pantopoulos http://orcid.org/0000-0002-2305-0057

## Ethics

All experimental procedures were approved by the Animal Care Committee of McGill University (protocol 4966).

## Decision letter and Author response

Decision letter https://doi.org/10.7554/eLife.81332.sa1
Author response https://doi.org/10.7554/eLife.81332.sa2

## Additional files

### Supplementary files
- MDAR checklist
- Supplementary file 1. List of primers used for qPCR.

### Data availability
All data generated or analysed during this study are included in the manuscript and supporting file.

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
