## [Editor Report]

The authors present a manuscript aiming to understand how systemic iron overload counteracts the hypoferremic effects of a specific inflammatory stimulus, specifically focused on the role of mechanisms of ferroportin regulation to achieve hypoferremia during inflammation. This work is of interest to the community of researchers interested in the interaction of systemic iron regulation and inflammation and possibly ultimately clinicians managing iron disorders. This work also is of novel significance for translational purposes and could lead to the design of better therapeutics for iron related disorders and/or anemia of chronic inflammation. The current study demonstrates LPS and exogenous hepcidin can synergistically lead to hypoferremia even in iron overload conditions and provides data implicating ferroportin translation in contributing to the fully sequestering iron in cells involved in iron flows to induce hypoferremia.

---

## [Decision Letter]

**Decision letter after peer review:**

[Editors’ note: the authors submitted for reconsideration following the decision after peer review. What follows is the decision letter after the first round of review.]

Thank you for submitting the paper "Coordinate regulation of liver ferroportin degradation and de novo synthesis determines serum iron levels in mice" for consideration by *eLife*. Your article has been reviewed by 3 peer reviewers, one of whom is a member of our Board of Reviewing Editors, and the evaluation has been overseen by a Senior Editor. The reviewers have opted to remain anonymous.

Comments to the Authors:

We are sorry to say that, after consultation with the reviewers, we have decided that this work will not be considered further for publication by *eLife*.

The main shortcomings of the presented work include a nebulously defined problem that makes for difficulty understanding; lack of clarity whether hepatocytes or macrophages are the specific cellular targets of hepcidin and LPS driving hypoferremia; incomplete analysis of dosing and timing of synthetic hepcidin to determine whether there is a proportion of systemic iron to overcoming hepcidin resistance; and lack of important endpoints of signaling pathways to hepcidin and IRP-IRE related changes as part of a more complete transcriptional regulation in addition to the translational regulation of FPN. The later point is aimed at the lack of a mechanistic understanding behind increased FPN synthesis. A complete list of reviewer comments can be found below.

*Reviewer #1 (Recommendations for the authors):*

Charlebois et al. present a well written manuscript focused on exploring the mechanism underlying coregulation of hepcidin by iron and inflammation. While iron overload has previously been shown to prevent effective hypoferremic response to inflammation, the mechanisms thereof are incompletely understood. The current work presents a compelling story of the important finding that inflammation induced hepcidin expression can be made more efficient in inducing hypoferremia by coupling it with iron restriction or exogenous hepcidin. Furthermore, there is an interesting clue that ferroportin can be translationally regulated by LPS to enhance iron egress and prevent robust hypoferremia. However, the authors do not reconcile how this work can be implemented, whether manipulating the nuanced distribution of iron (without decreasing iron overload systemically) has significance, or whether this better understanding is important for management of infection in iron overloaded patients. Furthermore, a substantive part of the presented data is not central to the main story, and in some cases, the conclusions are overstated. In addition, a more robust evaluation of signaling pathways in the liver in response to LPS is missing and important. Finally, it is not clear whether hypoferremia would be expected to decrease susceptibility to pathogens.

Charlebois et al. present a well written manuscript focused on exploring the mechanism underlying coregulation of hepcidin by iron and inflammation. While iron overload has previously been shown to prevent effective hypoferremic response to inflammation, the mechanisms thereof are incompletely understood. The current work presents a compelling story of the important finding that inflammation induced hepcidin expression can be made more efficient in inducing hypoferremia by coupling it with iron restriction or exogenous hepcidin. Furthermore, there is an interesting clue that ferroportin can be translationally regulated by LPS to enhance iron egress and prevent robust hypoferremia. However, the authors do not reconcile how this work can be implemented, whether manipulating the nuanced distribution of iron (without decreasing iron overload systemically) has significance, or whether this better understanding is important for management of infection in iron overloaded patients. Furthermore, a substantive part of the presented data is not central to the main story, and in some cases, the conclusions are overstated. In addition, a more robust evaluation of signaling pathways in the liver in response to LPS is missing and important. Finally, it is not clear whether hypoferremia would be expected to decrease susceptibility to pathogens. Some additional questions and clarifications are delineated below.

1) The authors consider hypoferremia to be serum iron below that of WT untreated mice on standard iron diet. This is not well explained. Why was this definition selected? In some cases, a statistically significant decrease in serum iron in response to an intervention was not meaningful as per the authors unless it resulted in hypoferremia to a certain degree (e.g. Figure 2a-2b). However, this is an unrealistic expectation that in iron loaded conditions that the same dose of LPS or hepcidin mimetic would lead to a proportionally larger decrease in iron than if the starting ferremia was much lower at baseline. This requires some explanation and corroboration.

2) Serum hepcidin is induced in LPS treated iron loaded mice in Figure 1 but increased hepcidin did not prevent iron overload. The authors interpret this as a decrease in hepcidin responsiveness. However, it would be useful to evaluate the signaling to hepcidin regulation, namely SMAD1/5/8, STAT3, and MEK/ERK1/2.

3) It is somewhat unclear why the authors are comparing WT iron loaded mice with Hjv ko mice and Hjv ko on an iron deficient diet. The effects in vivo of these composite changes make for inadequate comparisons. Please clarify in the introduction to preemptively elucidate why this is a sound comparison.

4) The authors overstate the statistics (e.g. page 5, like 93: "slightly elevated" where there is no statistically significant changes) or call things "borderline" when they are not. Please edit throughout to make clear that differences that are not significant are unlikely to be "truly" different one from the other. The only way to make this argument is to increase the number of mice in the repeated analysis. For example, using qualifiers like "profoundly" suppressed (page 7, line 133) or "modestly" affected (page 7, line 134) is subjective when the statistics drive the argument making the "modest" effect without significant difference an overinterpretation of the results. Similarly, page 7, line 147 suggesting splenic FPN suppression "in all animals" when only HJV ko on SD is statistically significant is misleading.

5) The authors spend a substantial amount of time recapitulating the already known elements regarding Hjv ko mice relative to WT and the effects of synthetic hepcidin. This focus detracts from the main points and all data that is not relevant to the main point should be moved to the supplementary material sections. This mainly pertains to Figure 1, 2, and 4.

6) The most interesting and novel part of the manuscript is the effects on FPN translation in the setting of LPS treatment. In addition, the effect of LPS on expression of iron transporters and sensors is additionally novel and under-developed. Is there a precedent for this? Please expand in the Discussion section. The manuscript as a whole would benefit from refocusing effort toward the data in Figure 6 and 7.

7) The authors claim that the results in WB in Figure 3 and 5 are a result of hepatocyte FPN expression despite using liver homogenates for these analyses while simultaneously suggesting the FPN signal in IHC is strongest in Kupffer cells in the liver. This can be resolved by analyzing the 2 populations separately using well-established liver digestion techniques. Otherwise, the first few sentences on page 7 are difficult to interpret objectively.

8) Page 8, line 171, there is mention that synthetic hepcidin did not promote inflammation but no direct evidence of this is provided. Please add expression and signaling data for inflammatory endpoints in the liver.

9) The main argument appears obvious, that the higher the iron load in the system, the more hepcidin is needed to induce hypoferremia due to the additional compensatory mechanisms that protect cells from iron toxicity. This should be made very clear in the abstract, introduction, and discussion.

10) Figure 6 is interesting and novel. However, the polysome data is significant only for WT vs. Hjv ko which is likely multifactorial, not singularly the result of dietary iron loading as the authors suggest (page 10, line 207). Having a WT IDD control would be helpful here. In addition, the generalization that Hjv ko represents all forms of iron overload is an overstatement.

11) Unclear in Figure 6B and 6C how the specific statistical comparison is important. Is it noted because it is the only observed difference between groups in this experiment or does it have specific importance? Please clarify?

*Reviewer #2 (Recommendations for the authors):*

The value of this study is the investigation of the relative role of iron overload and inflammation in the regulation of iron intake and iron distribution. In particular, the Authors investigated if and how inflammation can lead to hypoferremia in presence of iron overload. They analyze the expression of the proteins hepcidin and ferroportin in normal and HJV-KO animals (which are iron overloaded) in presence of administration of LPS, which triggers inflammation and increased expression of hepcidin. They showed that ferroportin is expressed at high levels under condition of iron overload and that administration of synthetic hepcidin is insufficient to trigger hypoferremia in iron-loaded animals. Hypoferremia was eventually achieved when LPS and synthetic hepcidin were administered concurrently. This can lead to design better therapeutics for iron related disorders.

The paper is well written and clear. The primary claims of the Authors are supported by their data, although some additional experiment are required to complete these studies.

Figure 7: The treatment with synthetic hepcidin (SH) is done only for one day. Although only the combination of LPS and SH shows a reduction in serum iron, long term affects were not evaluated. I think that a complete analysis of this approach should include longer studies to assess the effect on iron organ concentration comparing SH or LPS alone and the combination of the two reagents. Also, Figure 7 is not showing the effect of the various parameters using SH alone. (this data is somehow presented in figure 4, but it would be better compared the same animals side by side). I suspect that SH alone, in the long term, may also be sufficient to induce hypoferremia and reduce iron overload. Targeting ferroportin + SH is likely to induce hypoferremia and reduce iron overload faster than administration of SH alone.

In absence of appropriate mouse models, can the authors show, in a cellular model, that cells harboring the ferroportin gene in absence of the IRE sequence are less sensitive to export iron following iron administration, in presence or absence of SH?

*Reviewer #3 (Recommendations for the authors):*

The ability of animals to lower their extracellular iron concentration in response to inflammatory stimuli (hypoferremia of inflammation) is an important innate immune response that has been shown to mediate resistance to certain bacterial infections. Early studies of the molecular mechanism of hypoferremia noted that even mild iron overload, caused by excessive iron content of the common mouse diet, interfered with this response. The current study aimed to analyze the mechanism(s) behind this interference.

The strengths of the study include:

1) Two different models of iron overload were employed, a low hepcidin genetic model (Hjv-/-) and a high hepcidin iron-rich diet model (2% carbonyl iron) and a standard inflammatory stimulus 1 microgram/g of LPS. Both iron overload states greatly decreased the hypoferremic response, even though hepcidin was still induced by LPS.

2) The authors show that the mechanism of relative resistance to hepcidin is in large part caused by increased ferroportin translation in iron-overloaded tissues that counteracts the ferroportin-degrading and ferroportin-blocking effect of hepcidin.

3) The use of high-dose exogenous hepcidin as a probe of iron-overload-induced resistance to hepcidin is an important advance, with implications for the treatment of iron disorders.

The main weaknesses of the study include:

1) The paper is narrowly conceived and interpreted, making it mainly interesting to a very specialized readership. However, for this readership, the mechanistic insights are few and already anticipated.

2) The paper does not define the cell type that drives the hypoferremic response in each situation or mediates the resistance to hepcidin in iron overload (hepatocytes vs macrophages).

3) The evidence that iron overload also substantially raised ferroportin mRNA concentrations (Figure 2H), likely by a transcriptional mechanism, is not developed or interpreted in the context of the changes in translation.

4) Although the implication is that increased ferroportin translation in iron-overloaded tissues is mediated by the IRP-IRE system, and this is a reasonable hypothesis, the authors did not demonstrate this in their various models and conditions.

5) The paper provides strong evidence that iron overload induces a state of relative resistance to hepcidin, including hepcidin induced by LPS injections. Using phrases that explicitly state this would make the paper easier to understand.

6) Identifying the specific cellular targets of hepcidin and LPS (hepatocytes vs macrophages) that drive hypoferremia in the various conditions is not easy but it is important.

7) The induction of ferroportin by iron overload has a transcriptional component whose importance should be analyzed and commented on.

8) The role of IRE-IRP in the translational mechanism should be demonstrated.

9) A dose-response analysis of hepcidin resistance using exogenous hepcidin would be more convincing than a single dosage study.

[Editors’ note: further revisions were suggested prior to acceptance, as described below.]

Thank you for resubmitting your work entitled "A crosstalk between hepcidin and IRE/IRP pathways controls ferroportin expression and determines serum iron levels in mice" for further consideration by *eLife*. Your revised article has been evaluated by Mone Zaidi (Senior Editor) and a Reviewing Editor.

The manuscript has been improved but there are some remaining issues that need to be addressed, as outlined below:

The reviewers found merit in the work and appreciate the significant improvement in response to critiques of the original submission. However, the evaluation identified remaining weaknesses that prevent publication of this work in the present form. The main shortcomings of the presented work include a nebulously defined problem and suboptimal manuscript organization that makes for difficulty following, lack of clarity on the degree of effect of IRE mediated FPN expression, and the poor quality of the EMSA gels and IHC that hampers confirmation of the authors central conclusions. A complete list of reviewer comments can be found below. The requirements for further consideration would require the authors fully addressing the following:

1) Significant further edits to the manuscript to clarify the rationale, temper the conclusions, and organize the Results section with a more clear interpretation/meaning/shortcomings of results presented.

2) Because the EMSA gel and IHC are not quantitative and are the most important aspects underlying the central conclusions of the current work, either a more quantitative method to support the interpretation of these results or a modification of the conclusions is warranted.

*Reviewer #1 (Recommendations for the authors):*

Charlebois et al. present a well written and significantly improved manuscript focused on exploring the mechanism underlying coregulation of hepcidin by iron and inflammation. While iron overload has previously been shown to prevent effective hypoferremic response to inflammation, the mechanisms thereof are incompletely understood. The current work presents a compelling story of the important finding that inflammation induced hepcidin expression can be made more efficient in inducing hypoferremia by coupling it with iron restriction. Furthermore, there is new data demonstrating that iron counteracts the effects of induced hepcidin on post-translation ferroportin regulation by stabilizing Fpn(IRE+) mRNA to enhance its translation, thus enhancing iron egress and preventing robust hypoferremia in response to inflammation. The authors have added significant amounts of data and reorganized the manuscript to increase readability and support their conclusions. Some additional queries and clarifications remain.

1) Page 7, line 133-139, this paragraph appears out of place. It would be useful to understand why these analyses were done and the meaning of the results. For example, it appears that the authors intended "to evaluate compensatory effects on iron trafficking genes in response to LPS." Such a statement could be added to explain why these genes specifically were analyzed and what the results indicate at the end of this paragraph. Also, it is unclear why the authors are specifying Fpn(IRE+) here. Is the Fpn(IRE-) result not the same in response to LPS? These results should be presented all together. Please edit.

2) It would be much easier to read if the data comparing wt and Hjv-/- mice was presented separately from the data on LPS response. Again, much of the results on wt vs. Hjv-/- mice is already well established in the literature; as a consequence, only select comparisons are needed to support the specific points in LPS treated mice. Please consider moving all the wt vs Hjv-/- data to the supplement.

3) Page 8, line 157, "modestly affect it in wt mice (Figure 2B)" is stated while there is no statistically significant difference reported on the figure itself. Please edit.

4) Page 9, line 193-194, the authors report that synthetic hepcidin administration suppresses endogenous Hamp expression in the liver as a consequence of suppressed Tfr2(Figure 3G). However, it is not clear what the underlying mechanism would be. If it were just hypoferremia, Tfr2 concentration would be altered between SD and HID in wt mice and between wt and Hjv-/- mice and no such difference occurs. This is conjecture and not central to the major theme of this manuscript. Consider removing or moving to the supplement.

5) Figure 5A-5D require quantification as the loading controls are not homogeneous in most gels. Please edit to statistically corroborate claims. Also, high exposure gels do not add anything additional relative to low exposure gels. Please remove. In addition, there is still a problem with the non-parenchymal cells containing Kupffer cells as well as other cells and the claims should therefore be tempered as the findings are not specific for Kupffer cells. Finally, please add statistics to Figure 5E panel.

6) Figure 6A is unclear. What do the authors mean about shift from monosome to polysome? Where is that evident? May be better to break this single panel with many images into individual panels. This is very difficult to interpret as it is and the figure legend is consistently cumbersome, not providing sufficient clarity to enable effective interpretation of the details. The big picture message is that FPN(IRE+) is as expected modulated by iron status in the liver and thus would be expected to counteract hepcidin action on FPN in iron loading. However, as in Figure 5, the concept of whether the effect is in hepatocytes, Kupffer cells, or other non-parenchymal cells is not clarified. Please comment and provide additional study limitations on this point in the Discussion section.

7) What is the proportion of FPN(IRE+) vs FPN(IRE-) in hepatocytes and Kupffer cells? Has this previously been published? FPN(IRE+) in liver is more abundant that FPN(IRE-) but do we know if that is a consequence of changing the fraction of cell types in the liver? Is iron loading responsible for increasing the fraction of cells that express FPN(IRE+) or the amount of FPN(IRE+) expression per cell?

8) Figure 7 indicates that additional hepcidin can overcome the protective effect of iron overload to offset the effect presumably via an IRE-mediated increase in FPN. However, this is obvious and anticipated and provides only a correlation and not direct evidence supporting the IRE-mediated FPN compensation in inflammation hypothesis. This is also important to note in the discussion as a limitation of the study.

9) Second paragraph of the discussion (page14, line 304) and also later (page 16, line 342) continue to provide the expectation that hypoferremia is a binary phenomena. However, iron loaded mice also had a relative hypoferremia. This is important as there is no expectation that a higher starting point would be expected to reach the same nadir; this would require that the sensitivity/potency of the regulation was increased. This is not realistic. I would again urge the authors to more clearly define hypoferremia as a relative not absolute threshold concept and edit the discussion in line with this expectation.

*Reviewer #2 (Recommendations for the authors):*

Overall, the study could be potentially impactful, and the authors provided a large amount of data, but I found it difficult to follow as the questions were not clearly laid out, and it was not always clear what questions the experiments were trying to address. I think improving the logical flow of the writing would vastly improve the quality of the manuscript.

Figure 1 in particular was difficult to follow- the figure panels were not presented in order in the manuscript (eg, 1D was described after 1E and 1F). Some rewriting is in order to clarify this section.

While the authors carried out experiments on WT mice fed SD of HID, and Hjv -/- mice were fed SD or IDD, to achieve a broad spectrum of hepcidin regulation, it is difficult to make head to head comparisons between such a large number of variable and draw conclusions. "NTBI levels …seemed to decrease in Hjv-/- mice on IDD" -compared to what? This was not clear.

Perls staining in Figure 1F - please use higher magnification- it's difficult to draw conclusions from these images.

P.8 (line 169) In this assay LPS appeared to suppress splenic ferroportin -please state p value.

p.9 (line 187), Strikingly, hepcidin administration was much more effective in relatively iron depleted Hjv -/- mice on IDD and lowered serum iron and transferrin saturation below baseline - I'm not sure I agree with the use of the word "strikingly" as these mice are iron deficient?

Figure 3F, 3G and line 192-194 - synthetic hepcidin led to reduction of Hamp mRNA in WT mice on SD, possibly related to destabilization of Tfr2.

However, there is an even more drastic decrease in Tfr2 in hepcidin treated mice in Hjv -/- mice on IDD but no change in Hamp mRNA-can the authors comment?

Figure 4A - The authors draw conclusions on ferroportin localization based on IHC - IHC is not a quantitative method and the authors should find some other more quantitative method.

Line 210 "substantially reduced it (ferroportin) in Hjv -/- mice to almost control levels" -what is the control?

Figure 5 - given the quality of the EMSA gels, I'm not sure that the gels are quantitative (the lanes are blending into each other). It's very difficult to draw conclusions on IRP2.

*Reviewer #3 (Recommendations for the authors):*

The authors have properly responded to most issues raised by the reviewers but some of them remained unsolved. However, the data shown on FPN regulation rather suggest transcriptional regulation as a major driving forces (Figure 1) whereas IRP mediated regulation appears to be only of marginal importance. This needs to be clarified.

In addition, the use of different iron diets in wt and Hjv-/- mice still needs more explanation (for that rationale and in regard to the interpretation of the data) because different effects may be induced by iron availability independent of hepcidin signaling and Hjv functionality.

In the discussion the relevance of that finding for iron regulation in the setting of inflammation should be better emphazised.

---

## [Author Response]

[Editors’ note: the authors resubmitted a revised version of the paper for consideration. What follows is the authors’ response to the first round of review.]

Reviewer #1 (Recommendations for the authors):Charlebois et al. present a well written manuscript focused on exploring the mechanism underlying coregulation of hepcidin by iron and inflammation. While iron overload has previously been shown to prevent effective hypoferremic response to inflammation, the mechanisms thereof are incompletely understood. The current work presents a compelling story of the important finding that inflammation induced hepcidin expression can be made more efficient in inducing hypoferremia by coupling it with iron restriction or exogenous hepcidin. Furthermore, there is an interesting clue that ferroportin can be translationally regulated by LPS to enhance iron egress and prevent robust hypoferremia. However, the authors do not reconcile how this work can be implemented, whether manipulating the nuanced distribution of iron (without decreasing iron overload systemically) has significance, or whether this better understanding is important for management of infection in iron overloaded patients. Furthermore, a substantive part of the presented data is not central to the main story, and in some cases, the conclusions are overstated. In addition, a more robust evaluation of signaling pathways in the liver in response to LPS is missing and important. Finally, it is not clear whether hypoferremia would be expected to decrease susceptibility to pathogens.Charlebois et al. present a well written manuscript focused on exploring the mechanism underlying coregulation of hepcidin by iron and inflammation. While iron overload has previously been shown to prevent effective hypoferremic response to inflammation, the mechanisms thereof are incompletely understood. The current work presents a compelling story of the important finding that inflammation induced hepcidin expression can be made more efficient in inducing hypoferremia by coupling it with iron restriction or exogenous hepcidin. Furthermore, there is an interesting clue that ferroportin can be translationally regulated by LPS to enhance iron egress and prevent robust hypoferremia. However, the authors do not reconcile how this work can be implemented, whether manipulating the nuanced distribution of iron (without decreasing iron overload systemically) has significance, or whether this better understanding is important for management of infection in iron overloaded patients. Furthermore, a substantive part of the presented data is not central to the main story, and in some cases, the conclusions are overstated. In addition, a more robust evaluation of signaling pathways in the liver in response to LPS is missing and important. Finally, it is not clear whether hypoferremia would be expected to decrease susceptibility to pathogens. Some additional questions and clarifications are delineated below.1) The authors consider hypoferremia to be serum iron below that of WT untreated mice on standard iron diet. This is not well explained. Why was this definition selected? In some cases, a statistically significant decrease in serum iron in response to an intervention was not meaningful as per the authors unless it resulted in hypoferremia to a certain degree (e.g. Figure 2a-2b). However, this is an unrealistic expectation that in iron loaded conditions that the same dose of LPS or hepcidin mimetic would lead to a proportionally larger decrease in iron than if the starting ferremia was much lower at baseline. This requires some explanation and corroboration.

We have reduced the use of the term “hypoferremia” as “serum iron below that of wt untreated mice on standard diet” and replaced it with other terms, such as “…decrease serum iron” (p. 5), “drop in serum iron” (p. 8 and p. 15). The term “lack of hypoferremic response” has also been replaced with “relatively high circulating iron levels” (p. 8).

2) Serum hepcidin is induced in LPS treated iron loaded mice in Figure 1 but increased hepcidin did not prevent iron overload. The authors interpret this as a decrease in hepcidin responsiveness. However, it would be useful to evaluate the signaling to hepcidin regulation, namely SMAD1/5/8, STAT3, and MEK/ERK1/2.

We have measured markers of BMP/SMAD and IL-6/STAT3 signaling, the major hepcidin-inducing pathways, under all experimental settings. The new data are shown in Figures 1H-I and S6B-C.

3) It is somewhat unclear why the authors are comparing WT iron loaded mice with Hjv ko mice and Hjv ko on an iron deficient diet. The effects in vivo of these composite changes make for inadequate comparisons. Please clarify in the introduction to preemptively elucidate why this is a sound comparison.

On p. 6 we added a sentence stating that these dietary manipulations aimed “to achieve a broad spectrum of hepcidin regulation”. In fact, wt mice on high-iron diet and Hjv-/- mice of iron-deficient diet exhibit dramatic differences in hepcidin expression and have comparable liver iron load.

4) The authors overstate the statistics (e.g. page 5, like 93: "slightly elevated" where there is no statistically significant changes) or call things "borderline" when they are not. Please edit throughout to make clear that differences that are not significant are unlikely to be "truly" different one from the other. The only way to make this argument is to increase the number of mice in the repeated analysis. For example, using qualifiers like "profoundly" suppressed (page 7, line 133) or "modestly" affected (page 7, line 134) is subjective when the statistics drive the argument making the "modest" effect without significant difference an overinterpretation of the results. Similarly, page 7, line 147 suggesting splenic FPN suppression "in all animals" when only HJV ko on SD is statistically significant is misleading.

We have modified extensively the manuscript and adhered to these suggestions.

5) The authors spend a substantial amount of time recapitulating the already known elements regarding Hjv ko mice relative to WT and the effects of synthetic hepcidin. This focus detracts from the main points and all data that is not relevant to the main point should be moved to the supplementary material sections. This mainly pertains to Figure 1, 2, and 4.

We have moved old Figure 1 and parts of old Figures 2 and 4 to the supplementary material section (Figures S1, S2, S6A-C).

6) The most interesting and novel part of the manuscript is the effects on FPN translation in the setting of LPS treatment. In addition, the effect of LPS on expression of iron transporters and sensors is additionally novel and under-developed. Is there a precedent for this? Please expand in the Discussion section. The manuscript as a whole would benefit from refocusing effort toward the data in Figure 6 and 7.

We have performed new experiments to address the role of the IRE/IRP system in translational regulation of ferroportin. The data are shown in new Figure 5 and discussed. The responses of some iron transporters to LPS are known. This is explained in the Discussion section and appropriate references are provided.

7) The authors claim that the results in WB in Figure 3 and 5 are a result of hepatocyte FPN expression despite using liver homogenates for these analyses while simultaneously suggesting the FPN signal in IHC is strongest in Kupffer cells in the liver. This can be resolved by analyzing the 2 populations separately using well-established liver digestion techniques. Otherwise, the first few sentences on page 7 are difficult to interpret objectively.

The proposed experiment has been performed and the data are shown in new Figure 5E.

8) Page 8, line 171, there is mention that synthetic hepcidin did not promote inflammation but no direct evidence of this is provided. Please add expression and signaling data for inflammatory endpoints in the liver.

The proposed experiment has been performed and the data are shown in new Figure S6B. Since Zip14 and Lcn2 are also inflammatory markers, the data in Figure SF-G provide additional support.

9) The main argument appears obvious, that the higher the iron load in the system, the more hepcidin is needed to induce hypoferremia due to the additional compensatory mechanisms that protect cells from iron toxicity. This should be made very clear in the abstract, introduction, and discussion.

We fully agree with this view and have changed the text accordingly, as requested (see for instance p. 4 and p. 16).

10) Figure 6 is interesting and novel. However, the polysome data is significant only for WT vs. Hjv ko which is likely multifactorial, not singularly the result of dietary iron loading as the authors suggest (page 10, line 207). Having a WT IDD control would be helpful here. In addition, the generalization that Hjv ko represents all forms of iron overload is an overstatement.

The IRE-binding experiments shown in new Figure 5 provide a framework to interpret the polysome profile data in Figure 6 We agree that statistical significance is not reached in the shift of Fpn mRNA to polysomes in wt mice fed a high iron diet; however, the trend is clear. We have performed 3 biological replicates of this technically challenging and laborious experiment and with the additional support of the IRE-binding experiments, we are confident that all results in Figure 6 are biologically significant. We have rephrased the text to avoid the generalization that Hjv ko represents all forms of iron overload.

11) Unclear in Figure 6B and 6C how the specific statistical comparison is important. Is it noted because it is the only observed difference between groups in this experiment or does it have specific importance? Please clarify?

We agree, the way the statistical analysis was presented was confusing and did not have any specific importance. This is fixed in the new Figure 6B and 6C.

Reviewer #2 (Recommendations for the authors):The value of this study is the investigation of the relative role of iron overload and inflammation in the regulation of iron intake and iron distribution. In particular, the Authors investigated if and how inflammation can lead to hypoferremia in presence of iron overload. They analyze the expression of the proteins hepcidin and ferroportin in normal and HJV-KO animals (which are iron overloaded) in presence of administration of LPS, which triggers inflammation and increased expression of hepcidin. They showed that ferroportin is expressed at high levels under condition of iron overload and that administration of synthetic hepcidin is insufficient to trigger hypoferremia in iron-loaded animals. Hypoferremia was eventually achieved when LPS and synthetic hepcidin were administered concurrently. This can lead to design better therapeutics for iron related disorders.The paper is well written and clear. The primary claims of the Authors are supported by their data, although some additional experiment are required to complete these studies.Figure 7: The treatment with synthetic hepcidin (SH) is done only for one day. Although only the combination of LPS and SH shows a reduction in serum iron, long term affects were not evaluated. I think that a complete analysis of this approach should include longer studies to assess the effect on iron organ concentration comparing SH or LPS alone and the combination of the two reagents.

We agree that the fact that treatments with synthetic hepcidin were done only for one day is a potential limitation. The suggestion to perform long-term treatments is excellent; however, the cost for these experiments is not permissive. Moreover, the plasma half-life of hepcidin is known to be short, and this notion has sparked the development of more potent hepcidin mimetics.

Also, Figure 7 is not showing the effect of the various parameters using SH alone. (this data is somehow presented in figure 4, but it would be better compared the same animals side by side).

We have generated and embedded here an alternative Figure 7 that includes the hepcidin data shown in new Figure 3 (old Figure 4). If the reviewer feels that this is more informative, we will replace the current Figure 7 with the alternative.

I suspect that SH alone, in the long term, may also be sufficient to induce hypoferremia and reduce iron overload. Targeting ferroportin + SH is likely to induce hypoferremia and reduce iron overload faster than administration of SH alone.

A stable hepcidin analogue would be an excellent tool to address the issue raised by the reviewer. In fact, in preliminary experiments we found that high doses a potent hepcidin mimetic drug can trigger effective hypoferremia in iron-loaded mice. Unfortunately, we cannot include these data here because they are not complete, and we have not received clearance yet. They will be part of a separate ongoing study. Nevertheless, we added in the Discussion the following sentence: “…it is reasonable to predict that the use of more potent hepcidin analogs will overcome the antagonistic effects of increased ferroportin mRNA translation under iron overload”.

In absence of appropriate mouse models, can the authors show, in a cellular model, that cells harboring the ferroportin gene in absence of the IRE sequence are less sensitive to export iron following iron administration, in presence or absence of SH?

This is a great suggestion that requires the generation of cell lines stably transfected with ferroportin with and without IRE. Because this approach would be time consuming and moreover, may have limited physiological relevance, we opted to focus on determining the IRP responses to dietary iron manipulations in the whole liver and spleen, as well as in isolated hepatocytes and liver non-parenchymal cells (new Figure 5).

Reviewer #3 (Recommendations for the authors):The ability of animals to lower their extracellular iron concentration in response to inflammatory stimuli (hypoferremia of inflammation) is an important innate immune response that has been shown to mediate resistance to certain bacterial infections. Early studies of the molecular mechanism of hypoferremia noted that even mild iron overload, caused by excessive iron content of the common mouse diet, interfered with this response. The current study aimed to analyze the mechanism(s) behind this interference.The strengths of the study include:1) Two different models of iron overload were employed, a low hepcidin genetic model (Hjv-/-) and a high hepcidin iron-rich diet model (2% carbonyl iron) and a standard inflammatory stimulus 1 microgram/g of LPS. Both iron overload states greatly decreased the hypoferremic response, even though hepcidin was still induced by LPS.2) The authors show that the mechanism of relative resistance to hepcidin is in large part caused by increased ferroportin translation in iron-overloaded tissues that counteracts the ferroportin-degrading and ferroportin-blocking effect of hepcidin.3) The use of high-dose exogenous hepcidin as a probe of iron-overload-induced resistance to hepcidin is an important advance, with implications for the treatment of iron disorders.The main weaknesses of the study include:1) The paper is narrowly conceived and interpreted, making it mainly interesting to a very specialized readership. However, for this readership, the mechanistic insights are few and already anticipated.

We believe that the new manuscript provides critical mechanistic insights that strengthen the conclusions and increase the quality of this work.

2) The evidence that iron overload also substantially raised ferroportin mRNA concentrations (Figure 2H), likely by a transcriptional mechanism, is not developed or interpreted in the context of the changes in translation.

This is a valid point, as the increase of ferroportin mRNA content in iron overload is consistent throughout the manuscript. We refer to iron-mediated transcriptional induction of ferroportin in the Introduction (p. 4), discuss the possibility in the context of our data on pages 7 and 13.

3) Although the implication is that increased ferroportin translation in iron-overloaded tissues is mediated by the IRP-IRE system, and this is a reasonable hypothesis, the authors did not demonstrate this in their various models and conditions.4) The paper provides strong evidence that iron overload induces a state of relative resistance to hepcidin, including hepcidin induced by LPS injections. Using phrases that explicitly state this would make the paper easier to understand.

This critical point is addressed in the new manuscript with the data in Figure 5.

5) Identifying the specific cellular targets of hepcidin and LPS (hepatocytes vs macrophages) that drive hypoferremia in the various conditions is not easy but it is important.

This critical issue is addressed with the experiments shown in new Figure 5D-E.

6) The induction of ferroportin by iron overload has a transcriptional component whose importance should be analyzed and commented on.

Please, see response to point 2.

7) The role of IRE-IRP in the translational mechanism should be demonstrated.

This critical issue is addressed with the experiments shown in new Figure 5A-D.

8) A dose-response analysis of hepcidin resistance using exogenous hepcidin would be more convincing than a single dosage study.

Please, see responses to point 1 and point 2 to reviewer 2.

[Editors’ note: what follows is the authors’ response to the second round of review.]

The manuscript has been improved but there are some remaining issues that need to be addressed, as outlined below:Reviewer #1 (Recommendations for the authors):Charlebois et al. present a well written and significantly improved manuscript focused on exploring the mechanism underlying coregulation of hepcidin by iron and inflammation. While iron overload has previously been shown to prevent effective hypoferremic response to inflammation, the mechanisms thereof are incompletely understood. The current work presents a compelling story of the important finding that inflammation induced hepcidin expression can be made more efficient in inducing hypoferremia by coupling it with iron restriction. Furthermore, there is new data demonstrating that iron counteracts the effects of induced hepcidin on post-translation ferroportin regulation by stabilizing Fpn(IRE+) mRNA to enhance its translation, thus enhancing iron egress and preventing robust hypoferremia in response to inflammation. The authors have added significant amounts of data and reorganized the manuscript to increase readability and support their conclusions. Some additional queries and clarifications remain.1) Page 7, line 133-139, this paragraph appears out of place. It would be useful to understand why these analyses were done and the meaning of the results. For example, it appears that the authors intended "to evaluate compensatory effects on iron trafficking genes in response to LPS." Such a statement could be added to explain why these genes specifically were analyzed and what the results indicate at the end of this paragraph. Also, it is unclear why the authors are specifying Fpn(IRE+) here. Is the Fpn(IRE-) result not the same in response to LPS? These results should be presented all together. Please edit.

We have extensively edited this paragraph providing a rational for the analysis, as well as a concluding sentence, as requested. In addition, we have added qPCR data on Fpn(-IRE) expression in new Figure 2K.

2) It would be much easier to read if the data comparing wt and Hjv-/- mice was presented separately from the data on LPS response. Again, much of the results on wt vs. Hjv-/- mice is already well established in the literature; as a consequence, only select comparisons are needed to support the specific points in LPS treated mice. Please consider moving all the wt vs Hjv-/- data to the supplement.

We have thoroughly considered this suggestion. However, we were not satisfied with the outcome of alternative presentations of the data. While we agree that comparisons between wt and Hjv-/- mice are redundant, we feel that it is important to show these data as reference to compare responses of these animals to dietary iron manipulations (without or with LPS or hepcidin treatment). Therefore, we opted to keep the original structure of the figures.

3) Page 8, line 157, "modestly affect it in wt mice (Figure 2B)" is stated while there is no statistically significant difference reported on the figure itself. Please edit.

We edited the text indicating the lack of statistical significance.

4) Page 9, line 193-194, the authors report that synthetic hepcidin administration suppresses endogenous Hamp expression in the liver as a consequence of suppressed Tfr2(Figure 3G). However, it is not clear what the underlying mechanism would be. If it were just hypoferremia, Tfr2 concentration would be altered between SD and HID in wt mice and between wt and Hjv-/- mice and no such difference occurs. This is conjecture and not central to the major theme of this manuscript. Consider removing or moving to the supplement.

We removed these data to the supplement, as suggested.

5) Figure 5A-5D require quantification as the loading controls are not homogeneous in most gels. Please edit to statistically corroborate claims. Also, high exposure gels do not add anything additional relative to low exposure gels. Please remove. In addition, there is still a problem with the non-parenchymal cells containing Kupffer cells as well as other cells and the claims should therefore be tempered as the findings are not specific for Kupffer cells. Finally, please add statistics to Figure 5E panel.

We have now quantified the data in Figure 6A-6D (new numbering) as suggested; the data are shown in the right panels. We think that showing high exposure gels is necessary to demonstrate induction of IRP2. We have considerable experience with EMSA, as indicated by our technical publication on the topic in JoVE (ref. 27). In this paper (attached at the end of this rebuttal letter), we analyzed liver and spleen extracts from wt, Hjv-/-, Irp1-/- and Irp2-/- mice, and show both short and long exposures of the gels to better visualize IRE/IRP1 and IRE/IRP2 interactions. Regarding the issue with non-parenchymal and Kupffer cells, we agree that some statements should be tempered. As suggested, we did that throughout the manuscript (see p. 4, p. 11 and p. 17). We have now added statistics to Figure 6E panel as requested.

6) Figure 6A is unclear. What do the authors mean about shift from monosome to polysome? Where is that evident? May be better to break this single panel with many images into individual panels. This is very difficult to interpret as it is and the figure legend is consistently cumbersome, not providing sufficient clarity to enable effective interpretation of the details. The big picture message is that FPN(IRE+) is as expected modulated by iron status in the liver and thus would be expected to counteract hepcidin action on FPN in iron loading. However, as in Figure 5, the concept of whether the effect is in hepatocytes, Kupffer cells, or other non-parenchymal cells is not clarified. Please comment and provide additional study limitations on this point in the Discussion section.

We have modified Figure 7A (new numbering) by breaking the single panel as suggested. In addition, we revised the Figure legend, to make the description clear. Polysome profiling is a standard technique in the mRNA translation field. The shift from monosomes to polysomes is indicated in the graphs and quantified in the right panels. We agree that the relative contribution of hepatocytes, Kupffer cells and other non-parenchymal cells has not been fully clarified. As suggested, we comment on that in a new paragraph in the Discussion, highlighting limitations of the study (p. 17).

7) What is the proportion of FPN(IRE+) vs FPN(IRE-) in hepatocytes and Kupffer cells? Has this previously been published? FPN(IRE+) in liver is more abundant that FPN(IRE-) but do we know if that is a consequence of changing the fraction of cell types in the liver? Is iron loading responsible for increasing the fraction of cells that express FPN(IRE+) or the amount of FPN(IRE+) expression per cell?

The proportion of Fpn(+IRE) and Fpn(-IRE) has not been analyzed in freshly isolated hepatocytes and Kupffer cells. However, it has been analyzed in whole liver, as well as in hepatoma and macrophage cell lines. In all these settings, Fpn(+IRE) was shown to be predominant (ref. 21); we have indicated this on p. 12. It appears that Fpn(-IRE) is only enriched in intestinal epithelial cells (ref. 21).

8) Figure 7 indicates that additional hepcidin can overcome the protective effect of iron overload to offset the effect presumably via an IRE-mediated increase in FPN. However, this is obvious and anticipated and provides only a correlation and not direct evidence supporting the IRE-mediated FPN compensation in inflammation hypothesis. This is also important to note in the discussion as a limitation of the study.

We comment on this in the new Discussion section highlighting limitations of the study (p. 17).

9) Second paragraph of the discussion (page14, line 304) and also later (page 16, line 342) continue to provide the expectation that hypoferremia is a binary phenomena. However, iron loaded mice also had a relative hypoferremia. This is important as there is no expectation that a higher starting point would be expected to reach the same nadir; this would require that the sensitivity/potency of the regulation was increased. This is not realistic. I would again urge the authors to more clearly define hypoferremia as a relative not absolute threshold concept and edit the discussion in line with this expectation.

We have modified the sentences as suggested. In addition, to clarify this point we added values of ratios of serum iron between untreated and LPS- or hepcidin-treated mice in Figures 2A and 4A.

Reviewer #2 (Recommendations for the authors):Overall, the study could be potentially impactful, and the authors provided a large amount of data, but I found it difficult to follow as the questions were not clearly laid out, and it was not always clear what questions the experiments were trying to address. I think improving the logical flow of the writing would vastly improve the quality of the manuscript.Figure 1 in particular was difficult to follow- the figure panels were not presented in order in the manuscript (eg, 1D was described after 1E and 1F). Some rewriting is in order to clarify this section.

We have reorganized Figure 2 (new numbering) and also modified the respective section in the text, as suggested. We also did the same for Figure 1.

While the authors carried out experiments on WT mice fed SD of HID, and Hjv -/- mice were fed SD or IDD, to achieve a broad spectrum of hepcidin regulation, it is difficult to make head to head comparisons between such a large number of variable and draw conclusions. "NTBI levels …seemed to decrease in Hjv-/- mice on IDD" -compared to what? This was not clear.

We agree that the experimental setting is complex, but we could not find a better way to present the data (see also response to Q2 of reviewer 1). On p. 6 we now specify that “NTBI levels… seemed to decrease in Hjv-/- mice following IDD intake”.

Perls staining in Figure 1F - please use higher magnification- it's difficult to draw conclusions from these images.

We added a higher magnification image to the new figure (now Figure 2E), and we moved the lower magnification image to Figure 2-supplement 1.

P.8 (line 169) In this assay LPS appeared to suppress splenic ferroportin -please state p value.

The p value is now added on Figure 3D.

p.9 (line 187), Strikingly, hepcidin administration was much more effective in relatively iron depleted Hjv -/- mice on IDD and lowered serum iron and transferrin saturation below baseline - I'm not sure I agree with the use of the word "strikingly" as these mice are iron deficient?

We replaced “strikingly” with “notably”.

Figure 3F, 3G and line 192-194 - synthetic hepcidin led to reduction of Hamp mRNA in WT mice on SD, possibly related to destabilization of Tfr2.However, there is an even more drastic decrease in Tfr2 in hepcidin treated mice in Hjv -/- mice on IDD but no change in Hamp mRNA-can the authors comment?

We removed these data to the supplement, as also suggested by reviewer 1. Hamp mRNA is already undetectable in Hjv-/- mice on IDD, therefore hepcidin treatment is not expected to have any effect.

Figure 4A - The authors draw conclusions on ferroportin localization based on IHC - IHC is not a quantitative method and the authors should find some other more quantitative method.

We agree that IHC is not quantitative. We analyzed by Western and quantified relative ferroportin expression in hepatocytes and non-parenchymal liver cells in Figure 6E. We discuss limitations of the study to fully address these issues on p. 17.

Line 210 "substantially reduced it (ferroportin) in Hjv -/- mice to almost control levels" -what is the control?

We modified the sentence as “substantially reduced it (ferroportin) in Hjv -/- mice to almost wt levels”.

Figure 5 - given the quality of the EMSA gels, I'm not sure that the gels are quantitative (the lanes are blending into each other). It's very difficult to draw conclusions on IRP2.

As indicated in our response to Q5 of reviewer 1, we have considerable experience with EMSA. IRE/IRP2 bands are more difficult to visualize and often require longer exposures of the gel. We attach our technical publication in JoVE, where we analyzed liver and spleen extracts from wt and Hjv-/- mice. We hope that the reviewers can appreciate that the quality of the EMSAs in this work is similar to that in the JoVE paper.

Reviewer #3 (Recommendations for the authors):The authors have properly responded to most issues raised by the reviewers but some of them remained unsolved. However, the data shown on FPN regulation rather suggest transcriptional regulation as a major driving forces (Figure 1) whereas IRP mediated regulation appears to be only of marginal importance. This needs to be clarified.

The possible role of iron-dependent transcriptional induction of ferroportin is mentioned throughout the text. In response to this comment, we added a sentence in the “limitations” section of the Discussion (p. 17) stating that “The possible role of iron-dependent transcriptional induction of ferroportin in counterbalancing hepcidin actions requires further clarification”.

In addition, the use of different iron diets in wt and Hjv-/- mice still needs more explanation (for that rationale and in regard to the interpretation of the data) because different effects may be induced by iron availability independent of hepcidin signaling and Hjv functionality.

We discuss this in the “limitations” section of the Discussion (p. 17).

In the discussion the relevance of that finding for iron regulation in the setting of inflammation should be better emphazised.

Again, we discuss on p. 17 as limitation of the study that “the physiological implications of translational regulation of ferroportin in the broader setting of inflammation and/or infection have not been explored”.